# Physics-Integrated Variational Autoencoders for Robust and Interpretable Generative Modeling

**Naoya Takeishi,   Alexandros Kalousis**
University of Applied Sciences and Arts Western Switzerland (HES-SO)
Geneva, Switzerland
{naoya.takeishi,alexandros.kalousis}@hesge.ch

## Abstract

Integrating physics models within machine learning models holds considerable promise toward learning robust models with improved interpretability and abilities to extrapolate. In this work, we focus on the integration of incomplete physics models into deep generative models. In particular, we introduce an architecture of variational autoencoders (VAEs) in which a part of the latent space is grounded by physics. A key technical challenge is to strike a balance between the incomplete physics and trainable components such as neural networks for ensuring that the physics part is used in a meaningful manner. To this end, we propose a regularized learning method that controls the effect of the trainable components and preserves the semantics of the physics-based latent variables as intended. We not only demonstrate generative performance improvements over a set of synthetic and real-world datasets, but we also show that we learn robust models that can consistently extrapolate beyond the training distribution in a meaningful manner. Moreover, we show that we can control the generative process in an interpretable manner.

## 1   Introduction

Data-driven modeling is often opposed to theory-driven modeling, yet their integration has also been recognized as an important approach called *gray-box* or *hybrid* modeling. In statistical machine learning, incorporation of mathematical models of physics (in a broad sense; including knowledge of biology, chemistry, economics, etc.) has also been attracting attention. Gray-box / hybrid modeling in machine learning holds considerable promise toward learning robust models with improved abilities to extrapolate beyond the distributions that they have been exposed to during training. Moreover, it can bring significant benefits in terms of model interpretability since parts of a model get semantically grounded to concrete domain knowledge.

A technical challenge in *deep* gray-box modeling is to ensure an appropriate use of physics models. A careless design of models and learning can lead to an erratic behavior of the components meant to represent physics (e.g., with erroneous estimation of physics parameters), and eventually, the overall model just learns to ignore them. This is particularly the case when we bring together simplified or imperfect physics models with highly expressive data-driven machine learning models such as deep neural networks. Such cases call for principled methods for striking an appropriate balance between physics and data-driven models to prevent the detrimental effects during learning.

Integration of physics models into machine learning has been considered in various contexts (see, e.g., [44, 40] and our Section 4), but most existing studies focus on prediction or forecasting tasks and are not directly applicable to other tasks. More importantly, the careful orchestration of physics-based and data-driven components have not necessarily been considered. A notable exception is Yin et al. [47], in which they proposed a method to regularize the action of trainable components of a hybrid

model of differential equations. Their method has been developed for dynamics forecasting with additive combinations of physics and trainable models, but application to other situations is not trivial.

In this work, we aim at the integration of incomplete physics models into deep generative models. While we focus on variational autoencoders (VAEs, [15, 29]), our idea is applicable to other models in principle. In our VAE, the decoder comprises physics-based models and trainable neural networks, and some of the latent variables are semantically grounded to the parameters of the physics models. Such a VAE, if appropriately trained, is by construction partly interpretable. Moreover, since it can by construction capture the underlying physics, it will be robust in out-of-distribution regime and exhibit meaningful extrapolation properties. We propose a regularized learning framework for ensuring the meaningful use of the physics models and the preservation of the semantics of the latent variables in the physics-integrated VAEs. We empirically demonstrate that our method can learn a model that exhibits better generalization, and more importantly, can extrapolate robustly in out-of-distribution regime. In addition, we show how the direct access to the physics-grounded latent variables allows us to alter properties of generation meaningfully and explore counterfactual scenarios.

## 2   Physics-integrated VAEs

We first describe the structure of VAEs we consider, which comprise physics models and machine learning models such as neural nets. We suppose that the physics models can be solved analytically or numerically with a reasonable cost, and the (approximate) solution is differentiable with regard to the quantities on which the solution depends. This assumption holds in most physics models known in practice, which come in different forms such as algebraic and differential equations. If there is no closed-form solution of algebraic equations, we can utilize differentiable optimizers [3] as a layer of the model. For differential equations, differentiable integrators [see, e.g., 7] will constitute a layer. Handling non-differentiable and/or overly-complex simulators remains an important open challenge.

### 2.1   Example

We start with an example to demonstrate the main concepts. Let us suppose that data comprise time-series of the angle of pendulums following an ordinary differential equation (ODE):

$$\underbrace{\mathrm{d}^2\vartheta(t)/\mathrm{d}t^2 + \omega^2 \sin\vartheta(t)}_{\text{given as prior knowledge, } f_\mathrm{P}} + \underbrace{\xi \mathrm{d}\vartheta(t)/\mathrm{d}t - u(t)}_{\text{to be learned by NN, } f_\mathrm{A}} = 0, \tag{1}$$

where $\vartheta$ is a pendulum's angle, and $\omega$, $\xi$, and $u$ are the pendulum's angular velocity, damping coefficient, and external force, respectively. We suppose that a data point $x$ is a sequence of $\vartheta(t)$, i.e., $x = [\vartheta(0) \ \vartheta(\Delta t) \ \cdots \ \vartheta((\tau-1)\Delta t)]^\mathsf{T} \in \mathbb{R}^\tau$ for some $\Delta t \in \mathbb{R}$ and $\tau \in \mathbb{N}$, where $\vartheta(t)$ denotes the solution of (1) with a particular configuration of $\omega$, $\xi$, and $u$. In this example, we learn a VAE on a dataset comprising such $x$'s with different configurations of $\omega$, $\xi$, and $u$.

Suppose that the first two terms of (1) are given as prior knowledge, i.e., we know that the governing equation should contain $f_\mathrm{P}(\vartheta, z_\mathrm{P}) := \ddot{\vartheta} + z_\mathrm{P}^2 \sin\vartheta$. We will use such prior knowledge, $f_\mathrm{P}$, by incorporating it in the decoder of a VAE that we will learn. Since $f_\mathrm{P}$ misses some effects of the true system (1), we complete it by augmenting the decoder with a neural network $f_\mathrm{A}(\vartheta, \boldsymbol{z}_\mathrm{A})$. The VAE's latent variable will have two parts, $z_\mathrm{P}$ and $\boldsymbol{z}_\mathrm{A}$, respectively linked to $f_\mathrm{P}$ and $f_\mathrm{A}$. On one hand, $\boldsymbol{z}_\mathrm{A}$ works as an ordinary VAE's latent variable since $f_\mathrm{A}$ is a neural net, and we suppose $\boldsymbol{z}_\mathrm{A} \in \mathbb{R}^d$, $p(\boldsymbol{z}_\mathrm{A}) := \mathcal{N}(\boldsymbol{0}, \boldsymbol{I})$. On the other hand, we semantically ground $z_\mathrm{P}$ to a physics parameter; in this case, $z_\mathrm{P} \in \mathbb{R}$ should work as pendulum's $\omega$. In summary, the augmented decoder here is $\mathbb{E}[x] = \mathrm{ODEsolve}_\vartheta \big[ f_\mathrm{P}(\vartheta(t), z_\mathrm{P}) + f_\mathrm{A}(\vartheta(t), \boldsymbol{z}_\mathrm{A}) = 0 \big]$, where $\mathrm{ODEsolve}_\vartheta$ denotes some differentiable solver of an ODE with regard to $\vartheta$. The encoder will have corresponding recognition networks for $z_\mathrm{P}$ and $\boldsymbol{z}_\mathrm{A}$. The situation in this example will be numerically examined in Section 5.1.

### 2.2   General formulation

We now present the concept of our physics-integrated VAEs in a general form. Note that our interest is not limited to the additive model combination nor ODEs. In fact, the general formulation below subsumes non-additive augmentation of various physics models. The notation introduced in this section will be used to explain the proposed regularized learning method later in Section 3.

For ease of discussion, we suppose that a VAE decoder comprises two parts: a physics-based model $f_P$ and a trainable auxiliary function $f_A$. More general cases, for example with multiple trainable functions $f_{A,1}, f_{A,2}, \ldots$ used in different ways, are handled in Appendix A.

### 2.2.1 Latent variables and priors

We consider two types of latent variables, $z_P \in \mathcal{Z}_P$ and $z_A \in \mathcal{Z}_A$, which respectively will be used in $f_P$ and $f_A$. The latent variables can be in any space, but for the sake of discussion, we suppose $\mathcal{Z}_P$ and $\mathcal{Z}_A$ are (subsets of) the Euclidean space and set their prior distribution as multivariate normal:

$$p(z_P) := \mathcal{N}(z_P \mid m_P, v_P^2 I) \quad \text{and} \quad p(z_A) := \mathcal{N}(z_A \mid 0, I), \tag{2}$$

where $m_P$ and $v_P^2$ are defined in accordance with prior knowledge of $f_P$'s parameters. Note that $z_P$ will be directly interpretable as they will be semantically grounded to the parameters of the physics model $f_P$; for example in Section 2.1, $z_P := \omega$ was the angular velocity of a pendulum.

### 2.2.2 Decoder

The decoder of a physics-integrated VAE comprises two types of functions[1], $f_P \colon \mathcal{Z}_P \to \mathcal{Y}_P$ and $f_A \colon \mathcal{Y}_P \times \mathcal{Z}_A \to \mathcal{Y}_A$. For notational convenience, we consider a functional $\mathcal{F}$ that evaluates $f_P$ and $f_A$, solves an equation if any, and finally gives observation $x \in \mathcal{X}$. $\mathcal{X}$ may be the space of sequences, images, and so on. Assuming Gaussian observation noise, we write the observation model as

$$p_\theta(x \mid z_P, z_A) := \mathcal{N}\big(x \mid \mathcal{F}[f_A, f_P; z_P, z_A], \Sigma_x\big), \tag{3}$$

where $z_A \in \mathcal{Z}_A$ and $z_P \in \mathcal{Z}_P$ are the arguments of $f_A$ and $f_P$, respectively. Note that $f_A$ and $f_P$ may have other arguments besides $z_A$ and $z_P$, respectively, but they are omitted for simplicity. We denote the set of trainable parameters of $f_A$ and $f_P$ (and $\Sigma_x$) by $\theta$, while $f_P$ may have no trainable global parameters other than $z_P$.

Let us see the semantics of the functional[2] $\mathcal{F}$ first in the light of the example of Section 2.1. Recall that there we considered the additive augmentation of ODE (as in [47] and other studies). It is subsumed by the expression (3) by setting $\mathcal{F}[f_A, f_P; z_P, z_A] := \text{ODEsolve}[f_P(z_P) + f_A(z_A) = 0]$. Let us generalize the idea. Our definition of the decoder in (3) allows not only additive augmentation of ODE but also broader range of architectures. The composition of $f_P$ and $f_A$ is not limited to be additive because we consider general composition of functions $f_A$ and $f_P$. Moreover, the form of the physics model is not limited to ODEs. We list some examples of the configuration:

- If equation $f_P = 0$ has a closed-form solution $S_{f_P} \in \mathcal{Y}_P$ (assuming that the solution space coincides with $\mathcal{Y}_P$, just for ease of discussion), then $\mathcal{F}$ is simply an evaluation of $f_A$, for example, $\mathcal{F}[f_P, f_A; z_A] := f_A(S_{f_P}, z_A)$.
- If an algebraic equation $f_P = 0$ or $f_A \circ f_P = 0$ has no closed-form solution, then $\mathcal{F}$ will have a differentiable optimizer, e.g., $\mathcal{F}[f_P, f_A] := f_A(\arg\min \|f_P\|^2)$ or $\mathcal{F} := \arg\min \|f_A \circ f_P\|^2$.
- $f_P = 0$ or $f_A \circ f_P = 0$ can be a stochastic differential equation (and $\mathcal{F}$ contains its solver), for which $z_P$ and/or $z_A$ would become a sequence encoding the realization of the process noise.

The role of $f_A$ can also be diverse; it can work not only as a complement of physics models inside equations, but also as correction of numerical errors of solvers or optimizers, downsampling or upsampling, and observables (e.g., from angle sequence to video of a pendulum).

### 2.2.3 Encoder

The encoder of a physics-integrated VAE accordingly comprises two parts: for posterior inference of $z_P$ and for that of $z_A$. We consider the following decomposition of the approximated posterior:

$$q_\psi(z_P, z_A \mid x) := q_\psi(z_A \mid x) q_\psi(z_P \mid x, z_A),$$

$$\text{where} \quad q_\psi(z_A \mid x) := \mathcal{N}\big(z_A \mid g_A(x), \Sigma_A\big), \quad q_\psi(z_P \mid x, z_A) := \mathcal{N}\big(z_P \mid g_P(x, z_A), \Sigma_P\big). \tag{4}$$

---

[1] The distinction between $f_P$ and $f_A$ depends on the origin of the functional forms (and not if trainable or not). The form of $f_P$ depends on physics' insight and thus fixed. On the other hand, the form of $f_A$ is determined only from utility as a function appoximator, and we can use whatever useful (e.g., feed-forward NNs, RNNs, etc.).

[2] It is natural to consider that $\mathcal{F}$ is a functional (and not a function) because we may need the access to the functions $f_A$ and $f_P$ themselves, rather than their pointwise values. For example, we need the full access to those functions when the decoder has an ODE solver with arbitrary initial condition.

$g_A \colon \mathcal{X} \to \mathcal{Z}_A$ and $g_P \colon \mathcal{X} \times \mathcal{Z}_A \to \mathcal{Z}_P$ are recognition networks. We denote the trainable parameters of $g_A$ and $g_P$ (and $\Sigma_A$ and $\Sigma_P$) as $\psi$. This particular dependency is for our regularization method in Section 3.2, where $g_P$ should first remove the information of $z_A$ from $x$ and then infer $z_P$.

## 2.3 Evidence lower bound

The VAE is to be learned as usual by maximizing the lower bound of the marginal log likelihood known as evidence lower bound (ELBO). In our case, it is straightforward to derive:

$$
\begin{aligned}
\mathrm{ELBO}(\theta, \psi; \boldsymbol{x}) = {} & \mathbb{E}_{q_\psi(\boldsymbol{z}_P, \boldsymbol{z}_A | \boldsymbol{x})} \log p_\theta(\boldsymbol{x} \mid \boldsymbol{z}_P, \boldsymbol{z}_A) \\
& - D_{\mathrm{KL}}\big[q_\psi(\boldsymbol{z}_A \mid \boldsymbol{x}) \,\|\, p(\boldsymbol{z}_A)\big] - \mathbb{E}_{q_\psi(\boldsymbol{z}_A | \boldsymbol{x})} D_{\mathrm{KL}}\big[q_\psi(\boldsymbol{z}_P \mid \boldsymbol{x}, \boldsymbol{z}_A) \,\|\, p(\boldsymbol{z}_P)\big].
\end{aligned}
\tag{5}
$$

# 3 Striking balance between physics and trainable models

We propose a regularized learning objective for physics-integrated VAEs. It comprises two types of regularizers. The first is for regularizing unnecessary flexibility of function approximators like neural networks and presented in Section 3.1. The second is for grounding encoder's output to physics parameters and presented in Section 3.2. The overall objective is summarized in Section 3.3.

## 3.1 Regularizing excess flexibility of trainable functions

If the trainable component of the physics-integrated VAE (i.e., $f_A$) has rich expression capability, as is often the case with deep neural networks, merely maximizing the ELBO in (5) provides no guarantee that the physics-based component (i.e., $f_P$) will be used in a meaningful manner; e.g., $f_P$ may just be ignored. We want to ensure that $f_A$ does not unnecessarily dominate the behavior of the entire model and that $f_P$ is not ignored. To this end, we borrow an idea from the *posterior predictive check* (PPC), a procedure to check the validity of a statistical model [see, e.g., 9]. Whereas the standard PPCs examine the discrepancy between distributions of a model and data, we compute the discrepancy between those of the model and its "physics-only" reduced version, for monitoring and balancing the contributions of parts of the model.

For the sake of argument, suppose that a given physics model $f_P$ is completely correct for given data. Then, the discrepancy between the original model and its "physics-only" reduced model (where $f_A$ is somehow invalidated) should be close to zero because the decoder of both the original model (with $f_P$ and $f_A$ working) and the reduced model (with only $f_P$ working) should coincide in an ideal limit with the true data-generating process. Even if $f_P$ captures only a part of the truth, the discrepancy should be kept small, if not zero, to ensure meaningful use of the physics models in the overall model.

The "physics-only" reduced model is created as follows. Recall that the original VAE is defined by Eqs. (3) and (4). We define the decoder of the reduced model by replacing $f_A \colon \mathcal{Y}_P \times \mathcal{Z}_A \to \mathcal{Y}_A$ of (3) with a *baseline function* $h_A \colon \mathcal{Y}_P \to \mathcal{Y}_A$. That is, the reduced observation model is

$$
p_{\theta^r}^r(\boldsymbol{x} \mid \boldsymbol{z}_P, \boldsymbol{z}_A) := \mathcal{N}\big(\boldsymbol{x} \mid \mathcal{F}[h_A, f_P; \boldsymbol{z}_P], \Sigma_x\big),
\tag{3r}
$$

where we omit $z_A$ from the argument of $\mathcal{F}$ because $h_A$ no longer takes it. We denote the set of the trainable parameters of such a model as $\theta^r := \theta \setminus \mathrm{param}(f_A) \cup \mathrm{param}(h_A)$. The corresponding encoder is defined as follows. Recall that in the original model, posterior distributions of both $z_P$ and $z_A$ are inferred in (4) and then used for reconstructing each input $x$ in (3). On the other hand, in the "physics-only" reduced model, $z_A$ is not referred to by (3r), which makes it less meaningful to place a particular posterior of $z_A$ for each $x$. Hence, we define the "physics-only" encoder by marginalizing out $z_A$ and using prior[3] $p(z_A)$ instead. That is, the reduced posterior is

$$
q_\psi^r(\boldsymbol{z}_A, \boldsymbol{z}_P \mid \boldsymbol{x}) := p(\boldsymbol{z}_A) \int q_\psi(\boldsymbol{z}_P, \boldsymbol{z}_A \mid \boldsymbol{x}) \mathrm{d}\boldsymbol{z}_A.
\tag{4r}
$$

Below we give a guideline for the choice of the baseline function, $h_A$:

- If the ranges of $f_P$ and $f_A$ are the same (i.e., $\mathcal{Y}_P = \mathcal{Y}_A$), then $h_A$ can be an identity function $h_A = \mathrm{Id}$. Note that in the additive case $f_A \circ f_P = f_P + f_{A'}$, where $f_{A'}$ is a trainable function, replacing $f_A$ with $h_A = \mathrm{Id}$ is equivalent to replacing $f_{A'}$ with $h_{A'} = 0$.

---

[3]It is just for defining $q_\psi^r$ on the common support with $q_\psi$. Any non-informative distributions of $z_A$ are fine.

- If $\mathcal{Y}_\mathrm{P} \neq \mathcal{Y}_\mathrm{A}$, then $h_\mathrm{A}$ can be a linear or affine map from $\mathcal{Y}_\mathrm{P}$ to $\mathcal{Y}_\mathrm{A}$. For example, if $\mathcal{Y}_\mathrm{P} = \mathbb{R}^{d_\mathrm{P}}$ and $\mathcal{Y}_\mathrm{A} = \mathbb{R}^{d_\mathrm{A}}$ ($d_\mathrm{P} \neq d_\mathrm{A}$), then we can set $h_\mathrm{A}(f_\mathrm{P}(\boldsymbol{z}_\mathrm{P})) = \boldsymbol{W} f_\mathrm{P}(\boldsymbol{z}_\mathrm{P})$ where $\boldsymbol{W} \in \mathbb{R}^{d_\mathrm{A} \times d_\mathrm{P}}$.

The idea is to minimize the discrepancy between the full model and the "physics-only" reduced model. In particular, we minimize the discrepancy between the posterior predictive distributions

$$D_\mathrm{KL}\big[p_{\theta,\psi}(\tilde{\boldsymbol{x}} \mid X) \,\|\, p^\mathrm{r}_{\theta^\mathrm{r},\psi}(\tilde{\boldsymbol{x}} \mid X)\big], \quad \text{where}$$

$$p_{\theta,\psi}(\tilde{\boldsymbol{x}} \mid X) = \int p_\theta(\tilde{\boldsymbol{x}} \mid \boldsymbol{z}_\mathrm{P}, \boldsymbol{z}_\mathrm{A}) q_\psi(\boldsymbol{z}_\mathrm{P}, \boldsymbol{z}_\mathrm{A} \mid \boldsymbol{x}) p_\mathrm{d}(\boldsymbol{x} \mid X) \mathrm{d}\boldsymbol{z}_\mathrm{P} \mathrm{d}\boldsymbol{z}_\mathrm{A} \mathrm{d}\boldsymbol{x}, \tag{6}$$

$$p^\mathrm{r}_{\theta^\mathrm{r},\psi}(\tilde{\boldsymbol{x}} \mid X) = \int p^\mathrm{r}_{\theta^\mathrm{r}}(\tilde{\boldsymbol{x}} \mid \boldsymbol{z}_\mathrm{P}, \boldsymbol{z}_\mathrm{A}) q^\mathrm{r}_\psi(\boldsymbol{z}_\mathrm{P}, \boldsymbol{z}_\mathrm{A} \mid \boldsymbol{x}) p_\mathrm{d}(\boldsymbol{x} \mid X) \mathrm{d}\boldsymbol{z}_\mathrm{P} \mathrm{d}\boldsymbol{z}_\mathrm{A} \mathrm{d}\boldsymbol{x}.$$

$p_\mathrm{d}(\boldsymbol{x} \mid X)$ is the empirical distribution with the support on data $X \coloneqq \{\boldsymbol{x}_1, \ldots, \boldsymbol{x}_n\}$. We use $\tilde{\boldsymbol{x}}$, instead of $\boldsymbol{x}$, just for avoiding notational confusion by clarifying the target of integral $\int \mathrm{d}\boldsymbol{x}$.

Unfortunately, analytically computing (6) is usually intractable. Hence, we take the following upper bound of (6) (a proof is in Appendix B, and further remarks are in Appendix C):

**Proposition 1.** *Let $p_\theta$ and $p^\mathrm{r}_\theta$ be the shorthand of $p_\theta(\tilde{\boldsymbol{x}} \mid \boldsymbol{z}_\mathrm{P}, \boldsymbol{z}_\mathrm{A})$ in (3) and $p^\mathrm{r}_{\theta^\mathrm{r}}(\tilde{\boldsymbol{x}} \mid \boldsymbol{z}_\mathrm{P}, \boldsymbol{z}_\mathrm{A})$ in (3r), respectively. Let $p_\mathrm{P}$ and $p_\mathrm{A}$ be some distributions of $\boldsymbol{z}_\mathrm{P}$ and $\boldsymbol{z}_\mathrm{A}$, e.g., $p(\boldsymbol{z}_\mathrm{P})$ and $p(\boldsymbol{z}_\mathrm{A})$ using the priors in (2), respectively. The KL divergence in (6) can be upper bounded as follows:*

$$D_\mathrm{KL}\big[p_{\theta,\psi}(\tilde{\boldsymbol{x}} \mid X) \,\|\, p^\mathrm{r}_{\theta^\mathrm{r},\psi}(\tilde{\boldsymbol{x}} \mid X)\big] \leq \mathbb{E}_{p_\mathrm{d}(\boldsymbol{x}|X)}\Big[\mathbb{E}_{q_\psi(\boldsymbol{z}_\mathrm{P}, \boldsymbol{z}_\mathrm{A}|\boldsymbol{x})} D_\mathrm{KL}[p_\theta \,\|\, p^\mathrm{r}_\theta]$$

$$+ D_\mathrm{KL}[q_\psi(\boldsymbol{z}_\mathrm{A} \mid \boldsymbol{x}) \,\|\, p_\mathrm{A}] + \mathbb{E}_{q_\psi(\boldsymbol{z}_\mathrm{A}|\boldsymbol{x})} D_\mathrm{KL}[q_\psi(\boldsymbol{z}_\mathrm{P} \mid \boldsymbol{z}_\mathrm{A}, \boldsymbol{x}) \,\|\, p_\mathrm{P}]\Big]. \tag{7}$$

**Definition 1.** Let us denote the upper bound (7) by $\mathbb{E}_{p_\mathrm{d}(\boldsymbol{x}|X)}\hat{D}(\theta, \mathrm{param}(h), \psi; \boldsymbol{x})$. The regularization for inhibiting unnecessary flexibility of trainable functions is defined as minimization of

$$R_\mathrm{PPC}(\theta, \mathrm{param}(h), \psi) \coloneqq \mathbb{E}_{p_\mathrm{d}(\boldsymbol{x}|X)}\hat{D}(\theta, \mathrm{param}(h), \psi; \boldsymbol{x}). \tag{8}$$

*Remark* 1. When multiple trainable functions are differently used in a model (e.g., inside *and* outside an equation solver), which is often the case in practice, the definition of $R_\mathrm{PPC}$ should be generalized to consider marginal contribution of every trainable function. See Appendix A.

### 3.2 Grounding physics encoder by physics-based data augmentation

Toward properly learning physics-integrated VAEs, minimizing $R_\mathrm{PPC}$ solely may not be enough because inferred $\boldsymbol{z}_\mathrm{P}$ may be still meaningless but makes $R_\mathrm{PPC}$ not that large (e.g., with solution of $f_\mathrm{P}$ fluctuating around the mean pattern of data), and then optimization may not be able to escape such local minima. Though it is difficult to avoid such a local solution perfectly, we can alleviate the situation by considering additional objectives to encourage a proper use of the physics.

The idea is to use the physics model as a source of information for data augmentation, which helps us to ground the output of the recognition network, $g_\mathrm{P}$ in (4), to the parameters of $f_\mathrm{P}$. We want to draw some $\boldsymbol{z}_\mathrm{P}$, feed it to the physics model $f_\mathrm{P}$ (and a solver if any), and use the generated signal as additional data during training. A technical challenge to this end is that because the physics model may be incomplete, the artificial signals from it and the real signals may have different natures. To compensate such difference, we arrange a particular functionality of the physics encoder, $g_\mathrm{P}$.

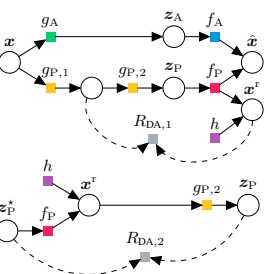

Let $\boldsymbol{z}^\star_\mathrm{P}$ be a sample drawn from some distribution of $\boldsymbol{z}_\mathrm{P}$ (e.g., prior $p(\boldsymbol{z}_\mathrm{P})$). We artificially generate signals $\boldsymbol{x}^\mathrm{r}(\boldsymbol{z}^\star_\mathrm{P})$ by feeding $\boldsymbol{z}^\star_\mathrm{P}$ to the "physics-only" decoding process in (3r), that is,

$$\boldsymbol{x}^\mathrm{r}(\boldsymbol{z}^\star_\mathrm{P}) \coloneqq \mathcal{F}[h_\mathrm{A}, f_\mathrm{P}; \boldsymbol{z}_\mathrm{P} = \boldsymbol{z}^\star_\mathrm{P}]. \tag{9}$$

We want the physics-part recognition network, $g_\mathrm{P}$, to successfully estimate $\boldsymbol{z}^\star_\mathrm{P}$ given the corresponding $\boldsymbol{x}^\mathrm{r}(\boldsymbol{z}^\star_\mathrm{P})$, which is necessary to say that the result of the inference by $g_\mathrm{P}$ is grounded to the parameters of $f_\mathrm{P}$. However, in general, real data $\boldsymbol{x}$ and the augmented data $\boldsymbol{x}^\mathrm{r}(\boldsymbol{z}^\star_\mathrm{P})$ have different natures because $f_\mathrm{P}$ may miss some aspects of the true data-generating process.

Figure 1: Diagrams of (*upper*) $R_\mathrm{DA,1}$ in (11) and (*lower*) $R_\mathrm{DA,2}$ in (12).

We handle this issue by considering a specific design of the physics-part recognition network, $g_P$. We decompose $g_P$ into two stages as $g_P(\boldsymbol{x}, \boldsymbol{z}_A) = g_{P,2}(g_{P,1}(\boldsymbol{x}, \boldsymbol{z}_A))$ without loss of generality. On one hand, $g_{P,1}$ should transform real data $\boldsymbol{x}$ to signals that resemble the physics-based augmented signal, $\boldsymbol{x}^r$. In other words, $g_{P,1}$ should "cleanse" real data into a virtual "physics-only" counterpart. We enforce such a functionality of $g_{P,1}$ by making its output close to the following quantity:

$$\boldsymbol{x}^r(g_P(\boldsymbol{x}, \boldsymbol{z}_A)) = \mathcal{F}[h_A, f_P; \boldsymbol{z}_P = g_P(\boldsymbol{x}, \boldsymbol{z}_A)]. \tag{10}$$

On the other hand, $g_{P,2}$ should receive such "cleansed" input and return the (sufficient statistics of) posterior of $\boldsymbol{z}_P$. If the aforementioned functionality of $g_{P,1}$ is successfully realized, we can directly self-supervise $g_{P,2}$ with $\boldsymbol{x}^r(\boldsymbol{z}_P^\star)$ because $\boldsymbol{x}^r(g_P(\boldsymbol{x}, \boldsymbol{z}_A))$ and $\boldsymbol{x}^r(\boldsymbol{z}_P^\star)$ should have similar nature.

In summary, we define a couple of regularizers for setting such functionality of $g_{P,1}$ and $g_{P,2}$ as follows (with the corresponding diagrams of computation shown in Figure 1):

**Definition 2.** Let $\mathrm{sg}[\cdot]$ be the stop-gradient operator. The regularization for the physics-based data augmentation is defined as minimization of

$$R_{\mathrm{DA},1}(\psi) := \mathbb{E}_{p_d(\boldsymbol{x}|X)q(\boldsymbol{z}_A|\boldsymbol{x})} \big\| g_{P,1}(\boldsymbol{x}, \boldsymbol{z}_A) - \mathrm{sg}\left[\boldsymbol{x}^r(g_P(\boldsymbol{x}, \boldsymbol{z}_A))\right] \big\|_2^2 \quad \text{and} \tag{11}$$

$$R_{\mathrm{DA},2}(\psi) := \mathbb{E}_{\boldsymbol{z}_P^\star} \big\| g_{P,2}\big(\mathrm{sg}\left[\boldsymbol{x}^r(\boldsymbol{z}_P^\star)\right]\big) - \boldsymbol{z}_P^\star \big\|_2^2. \tag{12}$$

### 3.3 Overall regularized learning objective

The overall regularized learning problem of the proposed physics-integrated VAEs is as follows:

$$\underset{\theta,\mathrm{param}(h),\psi}{\mathrm{minimize}} \quad - \mathbb{E}_{p_d(\boldsymbol{x}|X)}\mathrm{ELBO}(\theta, \psi; \boldsymbol{x}) + \alpha R_{\mathrm{PPC}}(\theta, \mathrm{param}(h), \psi) + \beta R_{\mathrm{DA},1}(\psi) + \gamma R_{\mathrm{DA},2}(\psi),$$

where each term appears in (5), (8), (11), and (12), respectively. Recall that $\theta$ and $\psi$ are the sets of the parameters of the full model's decoder (3) and encoder (4), respectively, and that $\mathrm{param}(h)$ denotes the set of the parameters of $h$, which may be empty. If we cannot specify a reasonable sampling distribution of $\boldsymbol{z}_P^\star$ needed in (12), we do not use $R_{\mathrm{DA},1}$ and $R_{\mathrm{DA},2}$; it may happen when the semantics of $\boldsymbol{z}_P$ are not inherently grounded, e.g., when $f_P$ is a *neural* Hamilton's equation [39].

## 4 Related work

The integration of theory-driven and data-driven methodologies has been sought in various ways. We overview some perspectives in this section and more in Appendix D.

**Physics+ML in model design** Integration in model design, often called gray-box or hybrid modeling, has been studied for decades [e.g., 24, 30, 38] and is still active, with deep neural networks utilized in various areas [e.g., 48, 27, 21, 41, 23, 1, 2, 8, 49, 42, 33, 16, 22, 5, 34, 26, 19, 25, 35]. Most recent studies focus on prediction, and the generative modeling has been less investigated. Moreover, mechanisms to regularize the flexibility of trainable components have hardly been addressed.

The work of Yin et al. [47] is notable here because they consider a mechanism to regularize the flexibility a trainable component to preserve the utility of physics in the model, even though it is only focused on dynamics learning for forecasting. They learn an additive hybrid ODE model $\dot{x} = f_P(x) + f_A(x)$, where $f_P$ is a prescribed physics model, and $f_A$ is a neural network. Such a model is subsumed in our architecture as exemplified in Section 2. Moreover, Yin et al. [47] propose to regularize $f_A$ by minimizing $\|f_A\|^2$. Such a term also appears in one of our regularizers, $R_{\mathrm{PPC}}$; when the observation noise is Gaussian, the first term of the right-hand side of (7) becomes $\mathbb{E}\|(f_A \circ f_P) - f_P\|_2^2 = \mathbb{E}\|f_P + f_{A'} - f_P\|_2^2 = \mathbb{E}\|f_{A'}\|_2^2$. Therefore, we get a "VAE variant" of Yin et al. [47] by switching off a part of $R_{\mathrm{PPC}}$ and the other regularizers, $R_{\mathrm{DA},1}$ and $R_{\mathrm{DA},2}$. We examine cases similar to it in our experiment for comparison.

Yıldız et al. [46] and Linial et al. [20] developed VAEs whose latent variable follows ODEs. Linial et al. [20] also suggest grounding the semantics of the latent variable by providing sparse supervision on it. It is feasible only when we have a chance to observe the latent variable (e.g., with an increased cost) and may often be inherently infeasible in some problem settings including ours. In our method, we never assume availability of observation of latent variables and instead use the physics models in a self-supervised manner. While direct comparison is not meaningful due to the difference of settings, we examine a baseline close to the base model of Linial et al. [20] in our experiment for comparison.

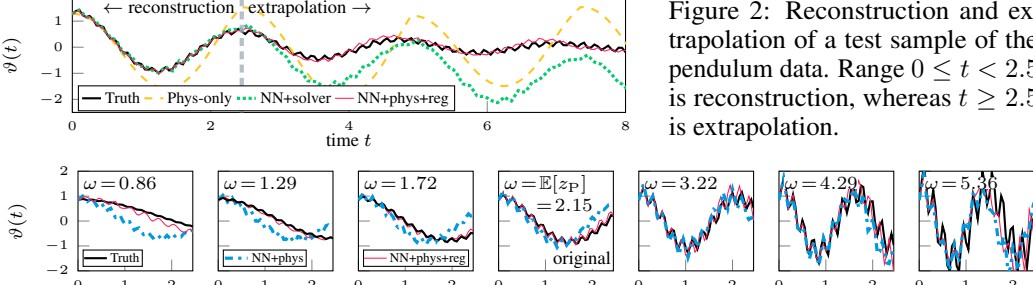

Figure 2: Reconstruction and extrapolation of a test sample of the pendulum data. Range $0 \leq t < 2.5$ is reconstruction, whereas $t \geq 2.5$ is extrapolation.

Figure 3: Counterfactual generation for the pendulum data. Horizontal axis is time $t$. The center panel shows the original data, and the rest is the generation with $z_P$ (i.e., $\omega$) altered while $\boldsymbol{z}_A$ fixed.

Toth et al. [39] propose a model where the latent variable sequence is governed by the Hamiltonian mechanics with a neural Hamiltonian. While it does not suppose very specific physics models but considers general mechanics, they can also be included in our framework; that is, $f_P$ can be a Hamilton's equation with a neural Hamiltonian. We try such a model in one of our experiments.

**Physics+ML in objective design**    Another prevailing strategy is to define objective functions based on physics knowledge [e.g., 36, 14, 28, 12, 45, 13, 50, 31, 6, 43]. In generative modeling, for example, Stinis et al. [37] use residuals from physics models as a feature of GAN's discriminator. Golany et al. [10] regularize the generation from GANs by forcing it close to a prescribed physics relation. These approaches are often easy to deploy, but an inherent limitation is that given physics knowledge should be complete to some extent, otherwise a physics-based loss is not well-defined.

## 5   Experiments

We performed experiments on two synthetic datasets and two real-world datasets, for which we prepared instances of physics-integrated VAEs. We show each particular architecture of physics-integrated VAEs and the corresponding results; some details are deferred to Appendix E. While direct comparison is impossible due to the differences of the problem settings, the baseline methods we examined (listed below) are similar to some existing methods [4, 46, 39, 20, 47].

| | |
|---|---|
| `NN-only` | Ordinary VAE [15, 29]; the decoder is $\mathbb{E}\boldsymbol{x} = f_A(\boldsymbol{z}_A)$, where $f_A$ is a neural net. |
| `Phys-only` | Physics VAE; the decoder is $\mathbb{E}\boldsymbol{x} = \mathcal{F}[f_P; \boldsymbol{z}_P]$ with no neural nets. The encoder is with neural nets as ordinary VAEs. This is almost equivalent to the method of Aragon-Calvo and Carvajal [4] when the problem is as in Section 5.3. |
| `NN+solver` | VAE with physics solvers; the decoder is $\mathbb{E}\boldsymbol{x} = \mathcal{F}[f_A; \boldsymbol{z}_A]$, where $f_A$ is a neural net, and $\mathcal{F}$ includes some equation-solving process (e.g., ODE/PDE solver), but no more physics-based knowledge is given (i.e., there is no $f_P$). This is similar to the methods of, for example, Yıldız et al. [46] and Toth et al. [39]. |
| `NN+phys` | Physics-integrated VAE learned without the regularizers (i.e., $\alpha = \beta = \gamma = 0$); this is similar to the base models of Linial et al. [20] and Qian et al. [25]. Finer ablations are also studied, among which the cases with $\beta = 0$ or $\gamma = 0$ are similar to the model of Yin et al. [47]. |
| `NN+phys+reg` | Our proposal; physics-integrated VAE learned with the proposed regularizers. |

We aligned the total dimensionality of the latent variables of each method (except `phys-only`); when $\dim \boldsymbol{z}_A = d_A$ and $\dim \boldsymbol{z}_P = d_P$ in `NN+phys(+reg)`, we set $\dim \boldsymbol{z}_A = d_A + d_P$ in `NN-only` and `NN+solver`. The hyperparameters, $\alpha$, $\beta$, and $\gamma$, were chosen with validation set performance. We investigated the performance sensitivity to them; no large degradation of performance was observed even if we changed the values by $\times 10$ or $\times \frac{1}{10}$ from the chosen values; details are in Appendix F.

### 5.1   Forced damped pendulum

**Dataset**    We generated data from (1) with $u(t) = A\omega^2 \cos(2\pi\phi t)$. Each data-point $\boldsymbol{x}$ is a sequence $\boldsymbol{x} := [\vartheta_1 \cdots \vartheta_\tau] \in \mathbb{R}^\tau$, where $\vartheta_j$ is the value of a solution $\vartheta(t_j)$ at $t_j := (j-1)\Delta t$. We randomly

Table 1: Reconstruction errors and inference errors on test sets of the pendulum data and the advection-diffusion data. Averages (and SDs) over 20 random trials are reported.

| | Pendulum | | | | Advection-diffusion | | | |
|---|---|---|---|---|---|---|---|---|
| | MAE of reconst. | | MAE of inferred $\omega$ | | MAE of reconst. | | MAE of inferred $a$ | |
| NN-only | 0.438 | $(2.9\times10^{-2})$ | – | | 0.0396 | $(2.2\times10^{-4})$ | – | |
| Phys-only | 1.55 | $(7.1\times10^{-4})$ | 0.232 | $(5.9\times10^{-3})$ | 0.393 | $(9.5\times10^{-4})$ | 0.0103 | $(1.5\times10^{-3})$ |
| NN+solver | 0.439 | $(2.3\times10^{-2})$ | – | | 0.0388 | $(1.7\times10^{-4})$ | – | |
| NN+phys | 0.370 | $(4.3\times10^{-2})$ | 1.04 | $(2.2\times10^{-1})$ | 0.0404 | $(1.2\times10^{-2})$ | 0.258 | $(3.2\times10^{-1})$ |
| NN+phys+reg | 0.363 | $(4.8\times10^{-2})$ | 0.229 | $(3.8\times10^{-2})$ | 0.0437 | $(1.5\times10^{-3})$ | 0.00951 | $(6.2\times10^{-3})$ |
| Ablations $\alpha=0$ | 0.396 | $(4.3\times10^{-2})$ | 0.889 | $(1.9\times10^{-1})$ | 0.0461 | $(1.3\times10^{-2})$ | 0.0444 | $(1.4\times10^{-2})$ |
| $\beta=0$ | 0.372 | $(4.1\times10^{-2})$ | 0.223 | $(3.6\times10^{-2})$ | 0.0747 | $(2.4\times10^{-2})$ | 0.199 | $(2.3\times10^{-1})$ |
| $\gamma=0$ | 0.381 | $(4.1\times10^{-2})$ | 0.276 | $(4.2\times10^{-2})$ | 0.0588 | $(9.1\times10^{-4})$ | 0.0548 | $(9.4\times10^{-7})$ |

drew a sample of the initial condition $\vartheta_1$ (with $\dot{\vartheta}_1 = 0$ fixed) and the values of $\omega$, $\zeta$, $A$, and $\phi$ for each sequence. We generated 2,500 sequences of length $\tau = 50$ with $\Delta t = 0.05$ and separated them into a training, validation, and test sets with 1,000, 500, and 1,000 sequences, respectively.

**Setting**   We set $f_\mathrm{P}$ as in Section 2.1, i.e., $f_\mathrm{P}(\vartheta, z_\mathrm{P}) := \ddot{\vartheta} + z_\mathrm{P}^2 \sin(\vartheta)$, where $z_\mathrm{P} \in \mathbb{R}$ should work as angular velocity $\omega$. We augmented it by $f_{\mathrm{A},1}(\vartheta, z_{\mathrm{A},1})$ additively, where $f_{\mathrm{A},1}$ was a multi-layer perceptron (MLP) and $z_{\mathrm{A},1} \in \mathbb{R}$. The ODE $f_\mathrm{P} + f_{\mathrm{A},1} = 0$ was solved with the Euler update scheme in the model. The model had another MLP[4] $f_{\mathrm{A},2}$ with another latent variable $\boldsymbol{z}_{\mathrm{A},2} \in \mathbb{R}^2$ for further modifying the solution of the ODE. In summary, the decoding process is $\mathcal{F} := f_{\mathrm{A},2}(\mathrm{solve}_\vartheta[f_\mathrm{P}(\vartheta, z_\mathrm{P}) + f_{\mathrm{A},1}(\vartheta, z_{\mathrm{A},1}) = 0], \boldsymbol{z}_{\mathrm{A},2})$. The construction of the proposed regularizer for such multiple $f_\mathrm{A}$'s is elaborated in Appendix A. We used $h_{\mathrm{A},1} = 0$ and $h_{\mathrm{A},2} = \mathrm{Id}$ as the baseline functions. The recognition networks, $g_{\mathrm{A},1}$, $g_{\mathrm{A},2}$, and $g_\mathrm{P}$, were modeled with MLPs. We used the initial element of each $\boldsymbol{x}$ as an estimation of the initial condition $\vartheta_1$.

**Results**   Figure 2 demonstrates a unique benefit of the hybrid modeling. We show an example of reconstruction with extrapolation. Recall that the training data comprise sequences of range $0 \le t < 2.5$ only; so the results in $t \ge 2.5$ are extrapolation (in time) rather than mere reconstruction. We can observe that while NN+solver cannot extrapolate even if it is equipped with an neural ODE, NN+phys+reg can reconstruct and extrapolate correctly.

Figure 3 illustrates well the advantage of the proposed regularizers. We show an example of generation from learned models with $z_\mathrm{P}$ manipulated. Recall that $z_\mathrm{P}$ is expected to work as pendulum's angular velocity $\omega$. We took a test sample with $\omega \approx \mathbb{E}[z_\mathrm{P}] \approx 2.15$ and generated signals with the original and different values of $z_\mathrm{P}$, keeping the values of $\boldsymbol{z}_\mathrm{A}$ to be the original posterior mean. We can see that the generation from NN+phys+reg matches better with the signals from the true process.

Table 1 (left half) summarizes the performance in terms of the reconstruction error and the inference error of physics parameter $\omega$ on the test set. The errors are reported in mean absolute errors (MAEs). The inference error of $\omega$ is evaluated by $|\mathbb{E}[z_\mathrm{P}] - \omega_\mathrm{true}|$. NN+phys+reg achieves small values in *both* reconstruction error and inference error. Meanwhile, the MAE of reconstruction by phys-only is significantly worse than those of the other methods, and the MAE of $\omega$ inferred by NN+phys is significantly worse than the others. These facts imply the effectiveness of the hybrid modeling and the proposed regularizers.

## 5.2   Advection-diffusion system

**Dataset**   We generated data from advection-diffusion PDE $\partial T/\partial t - a \cdot \partial^2 T/\partial s^2 + b \cdot \partial T/\partial s = 0$, where $s$ is the 1-D spatial dimension. We approximated the solution $T(s, t)$ on the 12-point even grid from $s = 0$ to $s = s_\mathrm{max}$, so each data-point $\boldsymbol{x}$ is a sequence of 12-dim vectors, i.e., $\boldsymbol{x} := [\boldsymbol{T}_1 \cdots \boldsymbol{T}_\tau] \in \mathbb{R}^{12\times\tau}$, where $\boldsymbol{T}_j := [T(0, t_j) \cdots T(s_\mathrm{max}, t_j)]^\mathsf{T}$ at $t_j := (j-1)\Delta t$. We set the boundary condition as $T(0, t) = T(s_\mathrm{max}, t) = 0$ and the initial condition as $T(s, 0) = c\sin(\pi s/s_\mathrm{max})$. We randomly drew $a$, $b$, and $c$ for each $\boldsymbol{x}$. We generated 2,500 sequences with $\tau = 50$ and $\Delta t = 0.02$ and separated them into a training, validation, and test sets with 1,000, 500, and 1,000 sequences, respectively.

---

[4]We used MLP as the data are fixed length. The same holds hereafter. Extension to other networks is easy.

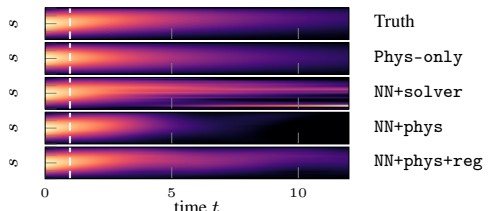
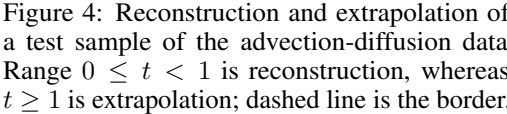

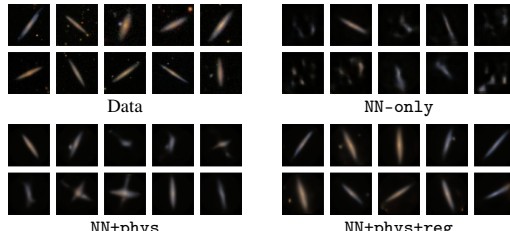

Figure 4: Reconstruction and extrapolation of a test sample of the advection-diffusion data. Range $0 \leq t < 1$ is reconstruction, whereas $t \geq 1$ is extrapolation; dashed line is the border.

Figure 5: (*upper left*) Subset of the galaxy image data. (*remaining*) Random generation from the learned models.

**Setting** We set $f_P$ as the diffusion PDE, i.e., $f_P(T, z_P) := \partial T/\partial t - z_P \partial^2 T/\partial s^2$, where $z_P \in \mathbb{R}$ should work as diffusion coefficient $a$. We augmented it by $f_A(T, \boldsymbol{z}_A)$ additively, where $f_A$ was an MLP and $\boldsymbol{z}_A \in \mathbb{R}^4$. Hence, the decoding process is $\mathcal{F} := \text{solve}_T[f_P(T, z_P) + f_A(T, \boldsymbol{z}_A) = 0]$. We used $h_A = 0$ as the baseline function. The recognition networks, $g_A$ and $g_P$, were modeled with MLPs. We used the initial snapshot of each sequence $\boldsymbol{x}$ as an estimation of the initial condition $\boldsymbol{T}_1$.

**Results** Figure 4 shows an example of reconstruction with extrapolation. As the training data only comprise sequences of range $0 \leq t < 1$, the remaining range $t \geq 1$ is extrapolation. Only NN+phys+reg (the bottom panel) achieves adequate extrapolation; phys-only lacks advection, NN+solver has unnatural artifacts, and NN+phys infers $z_P$ (i.e., diffusion coefficient $a$) wrongly.

Table 1 (right half) summarizes the reconstruction and inference errors, which are basically consistent with the results in the pendulum example, in the sense that NN+phys+reg achieves reasonable performance both in reconstruction and inference, while phys-only fails reconstruction, and NN+phys fails inference. Note that the reconstruction performance of NN+phys+reg is slightly worse than some baselines, which is probably due to suboptimal hyperparameters. In fact, with finer tuning of the hyperparameters, NN+phys+reg can achieve the reconstruction error closer to other methods while almost keeping the inference error[5]. We also show the performance of ablations of NN+phys+reg, where either of the regularizers was turned off (i.e., $\alpha = 0$, $\beta = 0$, or $\gamma = 0$). Not surprisingly their performance is worse than the full regularization, especially in terms of the inference error.

## 5.3 Galaxy images

**Dataset** We used images of galaxy of the Galaxy10 dataset [18]. We selected the 589 images of the "Disk, Edge-on, No Bulge" class and separated them into training, validation, and test sets with 400, 100, and 89 images, respectively. Each image is of size $69 \times 69$ with three channels. We performed data augmentation with random rotation and increased the size of the training set by 20 times.

**Setting** We set $f_P \colon \mathbb{R}^4_{>0} \to \mathbb{R}^{69 \times 69}$ as an exponential profile of the light distribution of galaxies [see 4, and references therein] whose input is $\boldsymbol{z}_P := [I_0 \ A \ B \ \vartheta]^\top \in \mathbb{R}^4_{>0}$. Let $[f_P(\boldsymbol{z}_P)]_{i,j}$ denote the $(i,j)$-element of the output of $f_P$. Then, for $1 \leq i, j \leq 69$, $[f_P(\boldsymbol{z}_P)]_{i,j} := I_0 \exp(-r_{i,j})$, where $r_{i,j}^2 := (\mathsf{X}_j \cos \vartheta - \mathsf{Y}_i \sin \vartheta)^2/A^2 + (\mathsf{X}_j \sin \vartheta + \mathsf{Y}_i \cos \vartheta)^2/B^2$, and $(\mathsf{X}_j, \mathsf{Y}_i)$ is the coordinate on the $69 \times 69$ even grid on $[-1, 1] \times [-1, 1]$. We modify the output of $f_P$ using a U-Net-like neural network $f_A \colon \mathbb{R}^{69 \times 69} \times \mathbb{R}^{\dim \boldsymbol{z}_A} \to \mathbb{R}^{69 \times 69 \times 3}$. Thus, the decoding process is $\mathcal{F} := f_A(f_P(\boldsymbol{z}_P), \boldsymbol{z}_A)$. We set $\dim \boldsymbol{z}_A = 2$ for NN+phys+reg. We set $h_A \colon \mathbb{R}^{69 \times 69} \to \mathbb{R}^{69 \times 69 \times 3}$ to be the repeat operator along the channel axis. The encoding process is as follows: first, features are extracted from an image $\boldsymbol{x}$ by a convolutional net like [4]. The extracted features are flattened and fed to MLPs $g_P$ and $g_A$.

---

[5]In the experiment with the advection-diffusion dataset reported in Table 1, the selected values of the hyperparameters were $\alpha = 0.1$, $\beta = 0.01$, and $\gamma = 10^6$, which were chosen from only eight candidates (see Appendix E for detail). When we instead set $\alpha = 0.032$, $\beta = 0.01$, and $\gamma = 10^6$ in the sensitivity experiment (shown in Appendix F), the reconstruction error of NN+phys+reg was 0.0390 ($4.5 \times 10^{-4}$), which is comparable to the baselines' performance in Table 1. In this setting, the inference error of NN+phys+reg was 0.0103 ($1.5 \times 10^{-3}$). We only reported the suboptimal values in Table 1 to align the granularity of the hyperparameter tuning grid with that in the experiment with the pendulum dataset.

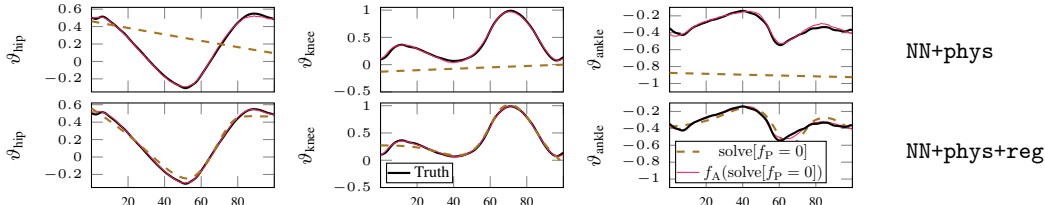

Figure 6: Reconstruction of a test sample of the gait data. Horizontal axis is normalized time.

**Results** Figure 5 shows an example of original data and random generation from the learned models. `NN-only` tends to generate non-realistic images, and `NN+phys` generates slightly better but still spuriously, whereas `NN+phys+reg` consistently generates galaxy-like images. More results (reconstruction, counterfactual generation, and inspection of latent variable) are deferred to Appendix F.

### 5.4 Human gait

**Dataset** We used a part of the dataset provided by [17], which contains measurements of locomotion at different speeds of 50 subjects. We extracted the angles of hip, knee, and ankle in the sagittal plane. Data originally comprise sequences of each stride normalized to be 100 steps, so each data-point $x$ is a sequence $\boldsymbol{x} := [\boldsymbol{\vartheta}_1 \cdots \boldsymbol{\vartheta}_{100}] \in \mathbb{R}^{3\times 100}$, where $\boldsymbol{\vartheta}_j := [\vartheta_{\text{hip},j}\ \vartheta_{\text{knee},j}\ \vartheta_{\text{ankle},j}]^\top$. We used different 400, 100, and 344 sequences as training, validation, and test sets, respectively.

**Setting** Biomechanical modeling of gait is a long-standing problem [see, e.g., 32]. We did not choose a specific model but let $f_{\text{P}}$ be a trainable Hamilton's equation as in [39, 11]. $\boldsymbol{z}_{\text{P}} \in \mathbb{R}^{2d_{\text{H}}}$ worked as the initial conditions of it, where $d_{\text{H}}$ was the dimensionality of the generalized position. We let $d_{\text{H}} = 3$ and modeled the neural Hamiltonian with an MLP. The solution of $f_{\text{P}} = 0$ was transformed by $f_{\text{A}}$ that also took $\boldsymbol{z}_{\text{A}} \in \mathbb{R}^{15}$ as an argument. In summary, the decoding process is $\mathcal{F} = f_{\text{A}}(\text{solve}[f_{\text{P}} = 0], \boldsymbol{z}_{\text{A}})$. We set $h_{\text{A}}$ to be an affine transform at each timestep, which had a weight matrix and a bias as $\text{param}(h)$. The recognition networks were modeled with MLPs.

**Results** Figure 6 is for visually comparing the difference of the learned models' behavior due to the proposed regularizers. We compare the reconstructions by `NN+phys` and `NN+phys+reg`. The dashed lines show an intermediate of the decoding process, i.e., $\text{solve}[f_{\text{P}} = 0]$, and the red solid lines show the final reconstruction, i.e., $f_{\text{A}}(\text{solve}[f_{\text{P}} = 0])$. Without the regularization (upper row), $\text{solve}[f_{\text{P}} = 0]$ returns almost meaningless signals, and $f_{\text{A}}$ bears the most effort of reconstruction. On the other hand, with the regularization (lower row), $\text{solve}[f_{\text{P}} = 0]$ already matches well the data, and $f_{\text{A}}$ modifies it only slightly. Superiority of the regularized model was also confirmed quantitatively; the average test reconstruction errors were 0.273 with `NN+phys` and 0.259 with `NN+phys+reg`.

## 6 Conclusion

Physics-integrated VAEs by construction attain partial interpretability as some of the latent variables are semantically grounded to the physics models, and thus we can generate signals in a controlled manner. Moreover, they have extrapolation capability due to the physics models. In this work, we proposed a regularized learning objective for ensuring a proper functionality of the integrated physics models. We empirically validated the aforementioned unique capability of physics-integrated VAEs and the importance of the proposed regularization method. In future studies, it would be interesting to investigate possibility and extension to learn a hybrid generative model with a highly complex observation process.

## Acknowledgments and Disclosure of Funding

This work was supported by the Innosuisse project *Industrial artificial intelligence for intelligent machines and manufacturing digitalization* (39453.1 IP-ICT) and the Swiss National Science Foundation Sinergia project *Modeling pathological gait resulting from motor impairments* (CRSII5_177179).

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
