# Appendix of
# "Physics-Integrated Variational Autoencoders for Robust and Interpretable Generative Modeling"

## Abstract

This document contains the appendix of the paper entitled "Physics-Integrated Variational Autoencoders for Robust and Interpretable Generative Modeling." Other supplementary materials include the codes and a part of the data we used in the experiments.

Appendix A provides a description of the physics-integrated VAEs and the proposed regularizers in a more general case than the one in the main text. Appendix B is dedicated to the proof of Proposition 1. Appendix C adds some remarks on the proposed regularizers. Appendix D additionally introduces related work as well as some examples of concrete architectures of physics-integrated models. Appendix E describes the experimental settings in detail. Appendix F contains the experimental results that did not fit in the main text. Finally, in Appendix G we discuss possible extensions of this work.

## A    General description of physics-integrated VAEs

In this section, we provide a general description of the physics-integrated VAEs and the proposed regularization method, since we only described a simple case in Sections 2 and 3 of the main text. The main difference of the general description from the simple one is the number of trainable function $f_{\mathrm{A}}$ in the model.

### A.1    Model

We here consider a generalized case in which we have multiple trainable models $f_{\mathrm{A},1}, f_{\mathrm{A},2}, \ldots, f_{\mathrm{A},K}$. We fix the number of $f_{\mathrm{P}}$ to be one as in the main text for clarity, while an extension in this regard is straightforward. We exemplify some use cases with multiple $f_{\mathrm{A}}$'s in Appendix D.

### A.1.1    Latent variables

Beside $\boldsymbol{z}_{\mathrm{P}} \in \mathcal{Z}_{\mathrm{P}}$, we consider $\boldsymbol{z}_{\mathrm{A},k} \in \mathcal{Z}_{\mathrm{A},k}$ for $k = 1, \ldots, K$. If $f_{\mathrm{A},k}$ does not take $\boldsymbol{z}$ as argument for some $k$, we simply suppose $\mathcal{Z}_{\mathrm{A},k} = \emptyset$ for such $k$. Otherwise, we suppose that $\mathcal{Z}_{\mathrm{A},k}$ is (some subset of) the Euclidean space for simplicity of discussion. The prior distributions are:

$$p(\boldsymbol{z}_{\mathrm{P}}) \coloneqq \mathcal{N}(\boldsymbol{z}_{\mathrm{P}} \mid \boldsymbol{m}_{\mathrm{P}}, v_{\mathrm{P}}^2 \boldsymbol{I}), \tag{A.1}$$

and

$$p(\boldsymbol{z}_{\mathrm{A},k}) \coloneqq \mathcal{N}(\boldsymbol{z}_{\mathrm{A},k} \mid \boldsymbol{0}, \boldsymbol{I}), \tag{A.2}$$

for $k$ whose $\mathcal{Z}_{\mathrm{A},k}$ is not empty.

35th Conference on Neural Information Processing Systems (NeurIPS 2021).

### A.1.2 Decoder

We intentionally do not specify the ranges and the domains of $f_\mathrm{P}$ and $f_{\mathrm{A},1}, f_{\mathrm{A},2}, \ldots, f_{\mathrm{A},K}$ because they depend on how these functions are connected each other. We denote the decoding process again with a functional $\mathcal{F}$ whose arguments are $f_\mathrm{P}$ and $f_{\mathrm{A},1}, \ldots, f_{\mathrm{A},K}$ as well as $\boldsymbol{z}$'s, that is, $\mathcal{F}[f_\mathrm{P}, f_{\mathrm{A},1}, \ldots, f_{\mathrm{A},K}; \boldsymbol{z}_\mathrm{P}, \boldsymbol{z}_{\mathrm{A},1}, \ldots, \boldsymbol{z}_{\mathrm{A},K}]$[1]. Inside $\mathcal{F}$ the functions can be connected in various ways; $\mathcal{F}$ can include 1) *in-equation* augmentation $\mathrm{solve}(f_\mathrm{P} + f_\mathrm{A} = 0)$ or $\mathrm{solve}(f_\mathrm{A} \circ f_\mathrm{P} = 0)$, 2) *out-equation* augmentation $f_\mathrm{A}(\mathrm{solve}(f_\mathrm{P} = 0))$, and 3) their arbitrary combinations, e.g., $f_{\mathrm{A},3}(\mathrm{solve}(f_{\mathrm{A},2}(f_\mathrm{P} + f_{\mathrm{A},1}) = 0))$. We show some examples in Appendix D. The observation model is

$$p_\theta(\boldsymbol{x} \mid \boldsymbol{z}_\mathrm{P}, \boldsymbol{z}_{\mathrm{A},1}, \ldots, \boldsymbol{z}_{\mathrm{A},K}) \coloneqq \mathcal{N}\big(\boldsymbol{x} \mid \mathcal{F}[f_\mathrm{P}, f_{\mathrm{A},1}, \ldots, f_{\mathrm{A},K}; \boldsymbol{z}_\mathrm{P}, \boldsymbol{z}_{\mathrm{A},1}, \ldots, \boldsymbol{z}_{\mathrm{A},K}], \boldsymbol{\Sigma}_x\big), \quad \text{(A.3)}$$

where $\theta$ is the set of trainable parameters of $f_\mathrm{P}$ and $f_{\mathrm{A},1}, \ldots, f_{\mathrm{A},K}$ (and $\boldsymbol{\Sigma}_x$).

### A.1.3 Encoder

Accordingly, the approximated posterior is

$$q_\psi(\boldsymbol{z}_\mathrm{P}, \boldsymbol{z}_{\mathrm{A},1}, \ldots, \boldsymbol{z}_{\mathrm{A},K} \mid \boldsymbol{x}) \coloneqq q_\psi(\boldsymbol{z}_{\mathrm{A},1}, \ldots, \boldsymbol{z}_{\mathrm{A},K} \mid \boldsymbol{x}) q_\psi(\boldsymbol{z}_\mathrm{P} \mid \boldsymbol{x}, \boldsymbol{z}_{\mathrm{A},1}, \ldots, \boldsymbol{z}_{\mathrm{A},K}). \quad \text{(A.4)}$$

We do not specify further structures of $q_\psi(\boldsymbol{z}_{\mathrm{A},1}, \ldots, \boldsymbol{z}_{\mathrm{A},K} \mid \boldsymbol{x})$ and $q_\psi(\boldsymbol{z}_\mathrm{P} \mid \boldsymbol{x}, \boldsymbol{z}_{\mathrm{A},1}, \ldots, \boldsymbol{z}_{\mathrm{A},K})$ because they depend on use cases. We denote the recognition networks for $\boldsymbol{z}_\mathrm{P}$ and $\boldsymbol{z}_{\mathrm{A},k}$ by $g_\mathrm{P}$ and $g_{\mathrm{A},k}$, respectively for $k = 1, \ldots, K$. $\psi$ is again the set of all the trainable parameters in the encoder side of the model.

### A.2 Regularizers

We slightly modify the definition of the proposed regularizers in accordance with the general description of the model.

The regularizer to suppress trainable components, $R_\mathrm{PPC}$, should be able to measure the contribution of all the trainable components, $f_{\mathrm{A},1}, \ldots, f_{\mathrm{A},K}$. While the original definition in Section 3 of the main text would still work as is, we empirically found that the following modification was useful in some cases. The idea is to consider the *marginal contribution* (compared to the physics model) of *each* of the trainable components, $f_{\mathrm{A},1}, \ldots, f_{\mathrm{A},K}$, instead of computing the contribution of all $f_\mathrm{A}$'s altogether. To show the essence of the idea, let us suppose $K = 2$. We consider the discrepancy between posterior predictive distributions for the following combinations:

$$D_{\mathrm{KL}}\big[p_{\theta,\psi}(\tilde{\boldsymbol{x}} \mid X) \,\|\, p_{\theta^\mathrm{r},\psi}^{\mathrm{r},\{1\}}(\tilde{\boldsymbol{x}} \mid X)\big], \quad \text{(A.5)}$$

$$D_{\mathrm{KL}}\big[p_{\theta,\psi}(\tilde{\boldsymbol{x}} \mid X) \,\|\, p_{\theta^\mathrm{r},\psi}^{\mathrm{r},\{2\}}(\tilde{\boldsymbol{x}} \mid X)\big], \quad \text{(A.6)}$$

$$D_{\mathrm{KL}}\big[p_{\theta^\mathrm{r},\psi}^{\mathrm{r},\{1\}}(\tilde{\boldsymbol{x}} \mid X) \,\|\, p_{\theta^\mathrm{r},\psi}^{\mathrm{r},\{1,2\}}(\tilde{\boldsymbol{x}} \mid X)\big], \quad \text{(A.7)}$$

$$D_{\mathrm{KL}}\big[p_{\theta^\mathrm{r},\psi}^{\mathrm{r},\{2\}}(\tilde{\boldsymbol{x}} \mid X) \,\|\, p_{\theta^\mathrm{r},\psi}^{\mathrm{r},\{1,2\}}(\tilde{\boldsymbol{x}} \mid X)\big], \quad \text{(A.8)}$$

where $p_{\theta^\mathrm{r},\psi}^{\mathrm{r},\mathcal{I}}(\tilde{\boldsymbol{x}} \mid X)$ ($\mathcal{I} \subseteq \{1, \ldots, K\}$) is a *partial* physics-only reduced model in which $f_{\mathrm{A},i}, \forall i \in \mathcal{I}$ are replaced with baseline function $h_{\mathrm{A},i}$. We let $p_{\theta^\mathrm{r},\psi}^{\mathrm{r},\mathcal{I}=\emptyset}(\tilde{\boldsymbol{x}} \mid X) \coloneqq p_{\theta,\psi}(\tilde{\boldsymbol{x}} \mid X)$ for convenience of notation.

Let us denote the upper bounds (see Proposition 1) of Eqs. (A.5)–(A.8) respectively as follows:

$$\mathbb{E}_{p_\mathrm{d}(\boldsymbol{x}|X)} \hat{D}_{\emptyset,\{1\}}(\theta, \mathrm{param}(h), \psi; \boldsymbol{x}),$$

$$\mathbb{E}_{p_\mathrm{d}(\boldsymbol{x}|X)} \hat{D}_{\emptyset,\{2\}}(\theta, \mathrm{param}(h), \psi; \boldsymbol{x}),$$

$$\mathbb{E}_{p_\mathrm{d}(\boldsymbol{x}|X)} \hat{D}_{\{1\},\{1,2\}}(\theta, \mathrm{param}(h), \psi; \boldsymbol{x}),$$

$$\mathbb{E}_{p_\mathrm{d}(\boldsymbol{x}|X)} \hat{D}_{\{2\},\{1,2\}}(\theta, \mathrm{param}(h), \psi; \boldsymbol{x}).$$

---

[1]Note that the expression in Section 2 of the main text, $\mathcal{F}[f_\mathrm{A}(f_\mathrm{P}(\boldsymbol{z}_\mathrm{P}), \boldsymbol{z}_\mathrm{A})]$, violates this general notation; for consistency, it should have been $\mathcal{F}[f_\mathrm{P}, f_\mathrm{A}; \boldsymbol{z}_\mathrm{P}, \boldsymbol{z}_\mathrm{A}]$ instead. The idea there was to emphasize the fact that $f_\mathrm{A}$ and $f_\mathrm{P}$ are *somehow* (not only additively) composited in the model.

Then, the regularizer is defined as

$$
\begin{aligned}
4R_{\mathrm{PPC}}&(\theta, \mathrm{param}(h), \psi) \\
&:= \mathbb{E}_{p_{\mathrm{d}}(\boldsymbol{x}|X)}\hat{D}_{\emptyset,\{1\}}(\theta, \mathrm{param}(h), \psi; \boldsymbol{x}) + \mathbb{E}_{p_{\mathrm{d}}(\boldsymbol{x}|X)}\hat{D}_{\emptyset,\{2\}}(\theta, \mathrm{param}(h), \psi; \boldsymbol{x}) \\
&\quad + \mathbb{E}_{p_{\mathrm{d}}(\boldsymbol{x}|X)}\hat{D}_{\{1\},\{1,2\}}(\theta, \mathrm{param}(h), \psi; \boldsymbol{x}) + \mathbb{E}_{p_{\mathrm{d}}(\boldsymbol{x}|X)}\hat{D}_{\{2\},\{1,2\}}(\theta, \mathrm{param}(h), \psi; \boldsymbol{x}).
\end{aligned}
\tag{A.9}
$$

The regularizer to use physics-based data augmentation, $R_{\mathrm{DA}}$, is defined in almost the same way as in the simple case — we draw samples $\boldsymbol{z}_{\mathrm{P}}^{\star}$ from some distribution of $\boldsymbol{z}_{\mathrm{P}}$ and generate physics-only augmentation by $\boldsymbol{x}^{\mathrm{r}}(\boldsymbol{z}_{\mathrm{P}}^{\star}) := \mathcal{F}[f_{\mathrm{P}}, h_{\mathrm{A},1}, \ldots, h_{\mathrm{A},K}; \boldsymbol{z}_{\mathrm{P}}^{\star}]$. Note that all of $f_{\mathrm{A}}$'s are replaced with $h_{\mathrm{A}}$'s at once unlike the aforementioned case of $R_{\mathrm{PPC}}$.

## B  Proof of Proposition 1

We use the following well-known facts in deriving the upper bound in Proposition 1.

**Lemma 1.** *Let $p_1(x, y)$ and $p_2(x, y)$ be two joint distributions on random variables $x$ and $y$, and $p_1(x)$ and $p_2(x)$ be the corresponding marginals. Then,*

$$
D_{\mathrm{KL}}[p_1(x) \parallel p_2(x)] \leq D_{\mathrm{KL}}[p_1(x, y) \parallel p_2(x, y)].
\tag{B.1}
$$

*Proof.* From definition,

$$
\begin{aligned}
D_{\mathrm{KL}}[p_1(x, y) \parallel p_2(x, y)] &= \int p_1(x, y) \frac{p_1(x, y)}{p_2(x, y)} \mathrm{d}x\mathrm{d}y \\
&= \int p_1(y \mid x) p_1(x) \frac{p_1(y \mid x) p_1(x)}{p_2(y \mid x) p_2(x)} \mathrm{d}x\mathrm{d}y \\
&= \int p_1(y \mid x) p_1(x) \frac{p_1(y \mid x)}{p_2(y \mid x)} \mathrm{d}x\mathrm{d}y + \int p_1(y \mid x) p_1(x) \frac{p_1(x)}{p_2(x)} \mathrm{d}x\mathrm{d}y \\
&= \int p_1(x) \left( \int p_1(y \mid x) \frac{p_1(y \mid x)}{p_2(y \mid x)} \mathrm{d}y \right) \mathrm{d}x + \int p_1(x) \frac{p_1(x)}{p_2(x)} \mathrm{d}x \\
&= \mathbb{E}_{p_1(x)} D_{\mathrm{KL}}[p_1(y \mid x) \parallel p_2(y \mid x)] + D_{\mathrm{KL}}[p_1(x) \parallel p_2(x)].
\end{aligned}
$$

Hence, from the nonnegativity of the KL divergence, we have

$$
\begin{aligned}
D_{\mathrm{KL}}[p_1(x) \parallel p_2(x)] &= D_{\mathrm{KL}}[p_1(x, y) \parallel p_2(x, y)] - \mathbb{E}_{p_1(x)} D_{\mathrm{KL}}[p_1(y \mid x) \parallel p_2(y \mid x)] \\
&\leq D_{\mathrm{KL}}[p_1(x, y) \parallel p_2(x, y)].
\end{aligned}
$$

$\square$

**Lemma 2.** *Let $x$ and $y$ be random variables with joint distribution $q(x, y)$. Let $I(x; y)$ be the mutual information between $x$ and $y$, i.e.: $I(x; y) := D_{\mathrm{KL}}[q(x, y) \parallel q(x)q(y)]$. Let $p(x)$ be some distribution of $x$. Then,*

$$
I(x; y) \leq \mathbb{E}_{q(y)} D_{\mathrm{KL}}\big[q(x \mid y) \parallel p(x)\big].
\tag{B.2}
$$

*Proof.* From the nonnegativity of the KL divergence,

$$
\begin{aligned}
I(x, y) &= D_{\mathrm{KL}}[q(x, y) \parallel q(x)q(y)] \\
&= \int q(x, y) \log \frac{q(x, y)}{q(x)q(y)} \mathrm{d}x\mathrm{d}y \\
&= \int q(x, y) \log \frac{q(x \mid y)}{q(x)} \mathrm{d}x\mathrm{d}y \\
&= \int q(x, y) \log \frac{q(x \mid y)p(x)}{p(x)q(x)} \mathrm{d}x\mathrm{d}y \\
&= \mathbb{E}_{q(y)} D_{\mathrm{KL}}\big[q(x \mid y) \parallel p(x)\big] - D_{\mathrm{KL}}\big[q(x) \parallel p(x)\big] \\
&\leq \mathbb{E}_{q(y)} D_{\mathrm{KL}}\big[q(x \mid y) \parallel p(x)\big].
\end{aligned}
$$

$\square$

Now we give a proof of Proposition 1.

*Proof of Proposition 1.* Let us denote the set of $\boldsymbol{z}_\mathrm{P}$ and $\boldsymbol{z}_\mathrm{A}$ by $z$. As a posterior predictive distribution $p(\tilde{\boldsymbol{x}} \mid X)$ is obtained by marginalizing out $z$ and $\boldsymbol{x}$ of joint distribution $p(\tilde{\boldsymbol{x}}, z, \boldsymbol{x} \mid X)$, from (B.1),

$$D_{\mathrm{KL}}\big[p_{\theta,\psi}(\tilde{\boldsymbol{x}} \mid X) \,\|\, p^{\mathrm{r}}_{\theta^{\mathrm{r}},\psi}(\tilde{\boldsymbol{x}} \mid X)\big] \leq D_{\mathrm{KL}}\big[p_{\theta,\psi}(\tilde{\boldsymbol{x}}, z, \boldsymbol{x} \mid X) \,\|\, p^{\mathrm{r}}_{\theta^{\mathrm{r}},\psi}(\tilde{\boldsymbol{x}}, z, \boldsymbol{x} \mid X)\big]. \qquad \text{(B.3)}$$

The right-hand side of (B.3) is

$$\begin{aligned}
&D_{\mathrm{KL}}\big[p_{\theta,\psi}(\tilde{\boldsymbol{x}}, z, \boldsymbol{x} \mid X) \,\|\, p^{\mathrm{r}}_{\theta^{\mathrm{r}},\psi}(\tilde{\boldsymbol{x}}, z, \boldsymbol{x} \mid X)\big] \\
&= D_{\mathrm{KL}}\Big[p_\theta(\tilde{\boldsymbol{x}} \mid z)q_\psi(z \mid \boldsymbol{x})p_{\mathrm{d}}(\boldsymbol{x} \mid X) \,\Big\|\, p^{\mathrm{r}}_{\theta^{\mathrm{r}}}(\tilde{\boldsymbol{x}} \mid z)q^{\mathrm{r}}_\psi(z \mid \boldsymbol{x})p_{\mathrm{d}}(\boldsymbol{x} \mid X)\Big] \\
&= \mathbb{E}_{p_{\mathrm{d}}(\boldsymbol{x}|X)}\mathbb{E}_{q_\psi(z|\boldsymbol{x})}D_{\mathrm{KL}}\big[p_\theta(\tilde{\boldsymbol{x}} \mid z) \,\|\, p^{\mathrm{r}}_{\theta^{\mathrm{r}}}(\tilde{\boldsymbol{x}} \mid z)\big] + \mathbb{E}_{p_{\mathrm{d}}(\boldsymbol{x}|X)}D_{\mathrm{KL}}\big[q_\psi(z \mid \boldsymbol{x}) \,\|\, q^{\mathrm{r}}_\psi(z \mid \boldsymbol{x})\big],
\end{aligned}$$

where the last term is

$$\begin{aligned}
&\mathbb{E}_{p_{\mathrm{d}}(\boldsymbol{x}|X)}D_{\mathrm{KL}}\big[q_\psi(z \mid \boldsymbol{x}) \,\|\, q^{\mathrm{r}}_\psi(z \mid \boldsymbol{x})\big] \\
&= \mathbb{E}_{p_{\mathrm{d}}(\boldsymbol{x}|X)}D_{\mathrm{KL}}\big[q_\psi(\boldsymbol{z}_\mathrm{P} \mid \boldsymbol{x}, \boldsymbol{z}_\mathrm{A})q_\psi(\boldsymbol{z}_\mathrm{A} \mid \boldsymbol{x}) \,\|\, q_\psi(\boldsymbol{z}_\mathrm{P} \mid \boldsymbol{x})p(\boldsymbol{z}_\mathrm{A})\big] \\
&= \mathbb{E}_{p_{\mathrm{d}}(\boldsymbol{x}|X)}\Big[\mathbb{E}_{q_\psi(\boldsymbol{z}_\mathrm{A}|\boldsymbol{x})}D_{\mathrm{KL}}\big[q_\psi(\boldsymbol{z}_\mathrm{P} \mid \boldsymbol{x}, \boldsymbol{z}_\mathrm{A}) \,\|\, q_\psi(\boldsymbol{z}_\mathrm{P} \mid \boldsymbol{x})\big] + D_{\mathrm{KL}}\big[q_\psi(\boldsymbol{z}_\mathrm{A} \mid \boldsymbol{x}) \,\|\, p(\boldsymbol{z}_\mathrm{A})\big]\Big] \\
&= \mathbb{E}_{p_{\mathrm{d}}(\boldsymbol{x}|X)}\Big[I(\boldsymbol{z}_\mathrm{P}; \boldsymbol{z}_\mathrm{A}) + D_{\mathrm{KL}}\big[q_\psi(\boldsymbol{z}_\mathrm{A} \mid \boldsymbol{x}) \,\|\, p(\boldsymbol{z}_\mathrm{A})\big]\Big].
\end{aligned}$$

Hence, from the upper bound of mutual information, (B.2), the right-hand side of (B.3) is further upper bounded as

$$\begin{aligned}
&D_{\mathrm{KL}}\big[p_{\theta,\psi}(\tilde{\boldsymbol{x}}, z, \boldsymbol{x} \mid X) \,\|\, p^{\mathrm{r}}_{\theta^{\mathrm{r}},\psi}(\tilde{\boldsymbol{x}}, z, \boldsymbol{x} \mid X)\big] \\
&\leq \mathbb{E}_{p_{\mathrm{d}}(\boldsymbol{x}|X)}\Big[\mathbb{E}_{q_\psi(z|\boldsymbol{x})}D_{\mathrm{KL}}\big[p_\theta(\tilde{\boldsymbol{x}} \mid z) \,\|\, p^{\mathrm{r}}_{\theta^{\mathrm{r}}}(\tilde{\boldsymbol{x}} \mid \boldsymbol{z}_\mathrm{P}, \boldsymbol{z}_\mathrm{A})\big] \\
&\qquad + \mathbb{E}_{q_\psi(\boldsymbol{z}_\mathrm{A}|\boldsymbol{x})}D_{\mathrm{KL}}\big[q_\psi(\boldsymbol{z}_\mathrm{P} \mid \boldsymbol{x}, \boldsymbol{z}_\mathrm{A}) \,\|\, p(\boldsymbol{z}_\mathrm{P})\big] + D_{\mathrm{KL}}\big[q_\psi(\boldsymbol{z}_\mathrm{A} \mid \boldsymbol{x}) \,\|\, p(\boldsymbol{z}_\mathrm{A})\big]\Big].
\end{aligned}$$

$\qquad\qquad\qquad\qquad\qquad\qquad\qquad\qquad\qquad\qquad\qquad\qquad\qquad\qquad\qquad\qquad\qquad\qquad\qquad\qquad$ $\square$

## C   Additional remarks on the regularized learning method

**Upper bound of KL in general case**   In the general case of Appendix A, the upper bound of the KL divergence used for defining $R_{\mathrm{PPC}}$ becomes slightly different. For example, a bound of (A.5) is as follows (recall that we focused the case of $K = 2$ for discussion):

$$\begin{aligned}
&D_{\mathrm{KL}}\big[p_{\theta,\psi}(\tilde{\boldsymbol{x}} \mid X) \,\|\, p^{\mathrm{r},\{1\}}_{\theta^{\mathrm{r}},\psi}(\tilde{\boldsymbol{x}} \mid X)\big] \leq \mathbb{E}_{p_{\mathrm{d}}(\boldsymbol{x}|X)}\Big[\mathbb{E}_{q_\psi(\boldsymbol{z}_\mathrm{P},\boldsymbol{z}_\mathrm{A}|\boldsymbol{x})}D_{\mathrm{KL}}[p_\theta \,\|\, p^{\mathrm{r},\{1\}}_\theta] \\
&+ D_{\mathrm{KL}}[q_\psi(\boldsymbol{z}_{\mathrm{A},1}, \boldsymbol{z}_{\mathrm{A},2} \mid \boldsymbol{x}) \,\|\, p_{\mathrm{A},\{1,2\}}] + \mathbb{E}_{q_\psi(\boldsymbol{z}_{\mathrm{A},1},\boldsymbol{z}_{\mathrm{A},2}|\boldsymbol{x})}D_{\mathrm{KL}}[q_\psi(\boldsymbol{z}_\mathrm{P} \mid \boldsymbol{z}_{\mathrm{A},1}, \boldsymbol{z}_{\mathrm{A},2}, \boldsymbol{x}) \,\|\, p_\mathrm{P}]\Big],
\end{aligned}$$

where $p_{\mathrm{A},\{1,2\}}$ is some distribution of $\boldsymbol{z}_{\mathrm{A},1}$ and $\boldsymbol{z}_{\mathrm{A},2}$, for example $p_{\mathrm{A},\{1,2\}} = p(\boldsymbol{z}_{\mathrm{A},1})p(\boldsymbol{z}_{\mathrm{A},2})$ using priors. This upper bound can be derived analogously to Proposition 1.

**Interpretation of upper bound**   It is interesting that the mutual information $I(\boldsymbol{z}_\mathrm{P}; \boldsymbol{z}_\mathrm{A})$ appears in the intermediate bound of $D_{\mathrm{KL}}\big[p_{\theta,\psi}(\tilde{\boldsymbol{x}} \mid X) \,\|\, p^{\mathrm{r}}_{\theta^{\mathrm{r}},\psi}(\tilde{\boldsymbol{x}} \mid X)\big]$ (see the proof of Proposition 1). Such a mutual information becomes a conditional mutual information (e.g., $I(\boldsymbol{z}_\mathrm{P}; \boldsymbol{z}_{\mathrm{A},1} \mid \boldsymbol{z}_{\mathrm{A},2})$) in the general case. Moreover, the last two terms of the upper bound in Proposition 1 are the same as the last two terms of the ELBO when $p_\mathrm{P}$ and $p_\mathrm{A}$ are the priors. In such a case, adding them as regularizers to the objective is equivalent to what is done in $\beta$-VAE [26]. It would also be interesting to discuss connection with the work by Zhao et al. [93].

**Usage of augmented data**   Data augmented with physics-based prior knowledge can also be used for pretraining (e.g., Jia et al. [28]). We rather generate and use them during the main training procedure as regularizers because the effects of pretraining may diminish in the main training.

# D   Related work

We introduce related studies that could not be in Section 4 of the main text due to length limit. Recall that in Section 4, we reviewed the studies with the following two perspectives: "Physics+ML in model design" and "Physics+ML in objective design." In this appendix, we follow a slightly different taxonomy: 1 ) *physics-integrated*, 2) *physics-informed*, and 3) *physics-inspired* methods. The first two of these three roughly correspond to the two perspectives in Section 4 of the main text. In contrast, we did not focus on the last one, physics-inspired method, in Section 4, while it will be informative for readers to provide a broader view of the context. We refer to some reviews and surveys on these topics, such as ones by Willard et al. [84], von Rueden et al. [79], von Rueden et al. [80], Beckh et al. [7], and Karniadakis et al. [33]. We would like to emphasize that the aforementioned three areas of research are never exclusive, and study that can bridge and unify them will be important.

## D.1   Physics-integrated methods

We refer to methods where the model is a combination of physics models and machine learning models as *physics-integrated*[2] ones. As such an approach was already explained to some extent in Section 4 of the main text, we here focus on exemplifying architectures of physics-integrated models. Most of the studies referred to here did not aim generative modeling originally, though the ideas can be fitted to our general architecture of physics-integrated VAEs. For more information, we recommend consulting the excellent survey / overview papers [e.g., 75, 34, 65, 63, 82, 10, 71, 84, 69].

**In-equation augmentation**   A numerical solver of dynamics models such as ODEs, PDEs, and discrete-time difference equations are one of the most prevailing forms of an equation-solving process that can be in a physics-integrated VAE. In such cases, $f_P$ and/or $f_A$ would give terms that appear in a dynamics equation. They are combined additively in many cases [66, 75, 64, 23, 50, 18, 67, 39, 88, 78, 51, 60], for example:

$$\mathcal{F} := \mathrm{solve}_y \left[ f_P(y, z_P) + f_A(y, z_A) = 0 \right], \tag{D.1}$$

where $\mathrm{solve}_y$ refers to a numerical ODE/PDE solver with regard to $y$ and returns the value of the solution on some time/space grid. Another way of combining $f_P$ and $f_A$ in this context is composition [59, 75, 45, 81, 38, 16, 52, 49, 8, 5, 42, 30, 36, 19], for example:

$$\mathcal{F} := \mathrm{solve}_y \left[ f_P(y, z_P, f_A(y, z_A)) = 0 \right], \tag{D.2}$$

where $f_A$ often gives estimation of some unknown or varying physics parameters in $f_P$. The order of the composition may reverse [recent examples include 1, 2], that is,

$$\mathcal{F} := \mathrm{solve}_y \left[ f_A(y, z_A, f_P(y, z_P)) = 0 \right], \tag{D.3}$$

where the output of a physics model is augmented by a machine learning model. Such a mechanism is often called *residual physics*. Some studies consider more complex combinations of $f_P$ and $f_A$, for example, $\mathcal{F} := \mathrm{solve}[f_{P,2}(f_A(f_{P,1})) = 0]$ [61, 17, 25, 31]. A trickier case appears in Jiang et al. [29], where discrete state of contact dynamics is first determined by a data-driven classifier, which is then used for choosing one of physics models (also including trainable ones) to be used. Moreover, Um et al. [77] considered to correct numerical errors by neural nets inside a differentiable solver of differential equations.

The equation-solving process can be anything else than an ODE/PDE solver. If (augmented) physics models are algebraic equations with closed-form solutions, $\mathcal{F}$ just evaluates some functions [e.g., 4]. If no closed-form solution is available, a diffentiable optimizer may be utilized in $\mathcal{F}$.

We also note that the latent force models [3] are known as a principled method to incorporate physics models in differential equations into Gaussian processes.

**Out-equation augmentation**   Physics and machine learning integration can also happen outside an equation-solving process. The simplest case is

$$\mathcal{F} := f_A(\mathrm{solve}[\cdots], z_A) \quad \text{or} \quad \mathcal{F} := f_A(\mathrm{solve}[\cdots], z_A) + \mathrm{solve}[\cdots], \tag{D.4}$$

where $\mathrm{solve}[\cdots]$ denotes the output of some equation-solving process, which also includes $f_P$ as well as another set of $f_A$'s. For example, such architectures can be found in the following use cases:

---

[2]Though this has been traditionally known as gray-box modeling, here we put an emphasis on the focus on physics-based models and adjust the wording with other related perspectives.

- $f_A$ corrects the output of an equation-solving process, $\text{solve}[\cdots]$, to compensate inaccuracy of physics models or unmodeled phenomena [89, 12, 55, 72, 82, 90, 58]. This can also be seen as residual physics.
- $f_A$ works as an observation function that changes signal's modality [24, 48, 87, 44, 76, 15, 68, 27].
- Output of $\text{solve}[\cdots]$ is used as input features of machine learning model $f_A$ [35, 57, 53, 92, 9, 58].

In [70], $f_A$ works as the weight of ensemble of physics models, that is,

$$\mathcal{F} := \sum_i f_{A,i}(\boldsymbol{z}_{A,i}) \cdot \text{solve}[\cdots]_i. \tag{D.5}$$

**Inverse problems as (V)AE**  The idea of (Bayesian) inverse problems is in line with the auto-encoding variational Bayes; in inverse problems, the forward process (i.e., a decoder) is known and a corresponding backward process (i.e., an encoder) is to be estimated. For example, Tait and Damoulas [74] propose a VAE whose decoder has a structure based on the finite element method for PDEs. Aragon-Calvo and Carvajal [4] replace VAE's decoder with a light distribution model of galaxies for inferring parameters of galaxy from images. Pakravan et al. [56] integrate a PDE solver into the decoder of a VAE. Nguyen and Bui-Thanh [54] discuss the form of solution for a special case where physics and VAEs are with linear models. Sun et al. [73] use learned surrogate models as the decoder of autoencoders. Similar problems are also discussed in the context of data assimilation [see, e.g., 22] and likelihood-free inference [see, e.g., 14].

### D.2  Physics-informed methods

We already introduced some studies in this direction, i.e., designing learning objective based on physics knowledge, in Section 4 of the main text. We call such an approach *physics-informed* after the work of Raissi et al. [62]. As it is not our main interest in this paper, we do not repeat the contents of Section 4; please refer to Section 4, and we also recommend consulting survey papers such as [33]. The study by Wang et al. [83] is also notable here as they analyze the difficulty of training physics-informed neural networks and propose a remedy.

### D.3  Physics-inspired methods

While the main interest of this work is integration of *application-specific* physics models into machine learning models, it is worth noting that there are lines of studies where the aim is to design models on the basis of *abstract and general* knowledge of data-generating process. The extent of the abstraction is diverse; in some studies, it is still natural to refer to the utilized knowledge as physics-related (in a narrow sense, i.e., as one of scientific disciplines) [13, 21, 32, 37, 43, 46, 47, 87, 39, 91], and in some other studies, the level of abstraction goes beyond that, e.g., a general model that can realize structural causal models is incorporated [40]. Hence, the heading of this subsection, *physics-inspired*, may not be perfect; we stick to it just for the consistency with the other perspectives.

For example, researchers have been investigating structured generative models for sequential data, in which the structure of latent variables reflects the sequential nature of data [13, 21, 32, 37, 43]. Moreover, Casale et al. [11] proposed to place a Gaussian process prior in VAEs. Note that these studies are never exclusive with the interest of our work and related ones; for example, the VAEs with sequential structures are indeed closely related to the VAEs with ODEs/PDEs [e.g., 87, 46, 47, 39, 91], since only the major difference is whether time is discrete or continuous. The techniques of the structured latent variable models would also be useful in physics-inspired and physics-integrated methods.

## E  Detailed experimental settings

### E.1  Infrastructure

We implemented the models using Python 3.8.0 with PyTorch 1.7.0 and NumPy 1.19.2 throughout the experiments. We used SciPy of version 1.5.2 in generating the synthetic datasets. The computation was performed with a machine equipped with an NVIDIA® Tesla™ V100 GPU in the experiment on the galaxy images dataset. We used a machine equipped with a CPU of Intel® Xeon® Gold 6148 in the other experiments.

### E.2 Forced damped pendulum

**Data-generating process**  We consider a gravity pendulum with damping effect and external force. Let $\vartheta(t)$ be the angle of the pendulum at time $t$. We generated the data by numerically integrating an ODE:

$$\frac{\mathrm{d}^2\vartheta(t)}{\mathrm{d}t^2} + \omega^2 \sin\vartheta(t) + \xi\frac{\mathrm{d}\vartheta(t)}{\mathrm{d}t} - A\omega^2\cos(2\pi\phi t) = 0,$$

using `scipy.integrate.solve_ivp` with the explicit Runge–Kutta method of order 8. The tolerance parameters `rtol` and `atol` were kept to be the default values, $10^{-3}$ and $10^{-6}$, respectively. We evaluated the solution's values at timesteps $t = 0, \Delta t, \cdots, (\tau - 1)\Delta t$ with $\Delta t = 0.05$ and $\tau = 50$ using the 7-th order interpolation polynomial. The values of the parameters, $\omega, \xi, A$, and $\phi$, as well as the initial condition $\vartheta(0)$ were randomly sampled when creating each sequence. The random sampling was with the uniform distributions on the following ranges: $\omega \in [0.785, 3.14], \xi \in [0, 0.8]$ $f \in [3.14, 6.28], A \in [0, 40]$, and $\vartheta(0) \in [-1.57, 1.57]$. The initial condition of $\dot\vartheta(0)$ was fixed to be 0. Each element of each generated sequence was added by zero-mean Gaussian noise with standard deviation 0.01.

**Data property**  The overall dataset we generated comprises 3,500 elements (data-points) in total. Each data-point $\boldsymbol{x}$ is a sequence of length $\tau$ of pendulum's angle, that is,

$$\boldsymbol{x}_i := [\vartheta_i(0)\ \vartheta_i(\Delta t)\ \cdots\ \vartheta_i((\tau - 1)\Delta t)]^\mathsf{T} \in \mathbb{R}^\tau,$$

where $i = 1, \ldots, 3500$ is the sample index.

**Train/valid/test split**  We first extracted 500 and 1,000 sequences randomly from the overall dataset as the validation set and the test set, respectively. We then selected 1,000 sequences out of the remaining 2,000 sequences to make a training set. This selection was randomly done every time; so a different random seed resulted in a different training set.

**Physics model**  A part of the data-generating process was given as physics model: $f_\mathrm{P}(\vartheta, z_\mathrm{P}) := \ddot\vartheta + z_\mathrm{P}\sin\vartheta$.

**Latent variables**  By construction of $f_\mathrm{P}$, $z_\mathrm{P} \in \mathbb{R}$ is expected to work in the same manner as $\omega$ in the data-generating process. There were also $\boldsymbol{z}_{\mathrm{A},1} \in \mathbb{R}$ and $\boldsymbol{z}_{\mathrm{A},2} \in \mathbb{R}^2$ in the full `NN+phys` and `NN+phys+reg` models. Meanwhile, we used $\boldsymbol{z}_{\mathrm{A},2} \in \mathbb{R}^4$ (and no $\boldsymbol{z}_{\mathrm{A},1}$, $z_\mathrm{P}$) in the `NN-only`; and $\boldsymbol{z}_{\mathrm{A},1} \in \mathbb{R}^2$ and $\boldsymbol{z}_{\mathrm{A},2} \in \mathbb{R}^2$ (and no $z_\mathrm{P}$) in the `NN+solver` model.

**Decoder architecture**  We describe the decoder architecture of the full `NN+phys` and `NN+phys+reg` models. In the first stage, an ODE $f_\mathrm{P}(\vartheta, z_\mathrm{P}) + f_{\mathrm{A},1}(\vartheta, \boldsymbol{z}_{\mathrm{A},1}) = 0$ is numerically solved with the Euler method for length $\tau$ with step size $\Delta t$. Let $\boldsymbol{\nu} \in \mathbb{R}^\tau$ be the solution sequence. In the second stage, $\boldsymbol{\nu}$ is then augmented by $f_{\mathrm{A},2}$, i.e., $f_{\mathrm{A},2}(\boldsymbol{\nu}, \boldsymbol{z}_{\mathrm{A},2})$. We modeled $f_{\mathrm{A},1}$ with a multilayer perceptron (MLP) with two hidden layers of size 64. We modeled $f_{\mathrm{A},2}$ also with an MLP with two hidden layers of size 128. We used the exponential linear unit (ELU) with its[3] $\alpha = 1.0$ as activation function after the hidden layers.

**Encoder architecture**  We describe the encoder architecture of the full `NN+phys` and `NN+phys+reg` models. We modeled the recognition networks, $g_{\mathrm{A},1}, g_{\mathrm{A},2}$, and $g_{\mathrm{P},2}$ with MLPs with five hidden layers of size 128, 128, 256, 64, and 32. We modeled $g_{\mathrm{P},1}$ as $g_{\mathrm{P},1}(\boldsymbol{x}, \boldsymbol{z}_{\mathrm{A},1}, \boldsymbol{z}_{\mathrm{A},2}) = \boldsymbol{x} + U(\boldsymbol{x}, \boldsymbol{z}_{\mathrm{A},1}, \boldsymbol{z}_{\mathrm{A},2})$, where $U$ was an MLP with two hidden layers of size 128. We used ELU with its[3] $\alpha = 1.0$ as activation function after the hidden layers. We put a softplus function after the final output of $g_\mathrm{P}$ to make its output positive-valued.

**Replacement functions**  To create the reduced models, we replaced $f_{\mathrm{A},1}$ and $f_{\mathrm{A},2}$ respectively by $h_{\mathrm{A},1} = 0$ and $h_{\mathrm{A},2} = \mathrm{Id}$.

---

[3] $\alpha$ here is different from one of the hyperparameters of the proposed regularizers.

**Hyperparameters**  We selected the hyperparameters, $\alpha$, $\beta$, and $\gamma$, from the following sets: $\alpha \in \{10^{-3}, 10^{-2}, 10^{-1}\}$, $\beta \in \{10^{-4}, 10^{-3}, 10^{-2}\}$, and $\gamma \in \{10^{-2}, 10^{-1}, 1\}$,. These ranges were chosen to roughly adjust the values of the corresponding regularizers to that of the ELBO. The configuration that achieved the best reconstruction error on the validation set was selected finally: $\alpha = 10^{-2}$, $\beta = 10^{-3}$, and $\gamma = 10^{-1}$. In computing $R_{\mathrm{DA},2}$, we sampled $z_{\mathrm{P}}^*$ from the uniform distribution on range $[0.392, 3.53]$.

**Optimization**  We used the Adam optimizer with its[4] $\alpha = 10^{-3}$, $\gamma_1 = 0.9$, $\gamma_2 = 0.999$, and $\epsilon = 10^{-3}$. We ran iterations with mini-batch size 200 for 5000 epochs (i.e., 25,000 iterations in total) and saved the model that achieved the best validation reconstruction error.

### E.3 Advection-diffusion system

**Data-generating process**  We consider the advection (convection) and diffusion of something (e.g., heat) on the 1-dimensional space, which is described by the following PDE:
$$\frac{\partial T(t,s)}{\partial t} - a\frac{\partial^2 T(t,s)}{\partial s^2} + b\frac{\partial T(t,s)}{\partial s} = 0,$$
where $t$ and $s$ denote the time and space dimension, respectively. We numerically solved this PDE using `scipy.integrate.solve_ivp` with the explicit Runge–Kutta method of order 8. The spatial derivative was computed with discretization on the $H$-point even grid between $s = 0$ and $s = s_{\max}$ with $H = 12$ and $s_{\max} = 2$. We evaluated the solutions values at timesteps $t = 0, \Delta t, \cdots, (\tau - 1)\Delta t$ with $\Delta t = 0.02$ and $\tau = 50$. The initial condition was set $T(0, s) = c\sin(\pi s/s_{\max})$, and we set the Dirichlet boundary condition $T(t, 0) = T(t, s_{\max}) = 0$. The values of the parameters $a$, $b$, and $c$ were randomly sampled when creating each sequence. The random sampling was with the uniform distributions on the following ranges: $a \in [10^{-2}, 10^{-1}]$, $b \in [10^{-2}, 10^{-1}]$, and $c \in [0.5, 1.5]$. Each element of each generated sequence was added by zero-mean Gaussian noise with standard deviation 0.001.

**Data property**  The overall dataset we generated comprises 3,500 sequences, each of which is
$$\boldsymbol{x}_i := \begin{bmatrix} T_i(0,0) & T_i(\Delta t, 0) & \cdots & T_i((\tau-1)\Delta t, 0) \\ \vdots & \vdots & & \vdots \\ T_i(0, s_{\max}) & T_i(\Delta t, s_{\max}) & \cdots & T_i((\tau-1)\Delta t, s_{\max}) \end{bmatrix} \in \mathbb{R}^{H \times \tau}.$$

**Train/valid/test split**  We first extracted 500 and 1,000 sequences randomly from the overall dataset as the validation set and the test set, respectively. We then selected 1,000 sequences out of the remaining 2,000 sequences to make a training set. This selection was randomly done every time; so a different random seed resulted in a different training set.

**Physics model**  A part of the data-generating process was given as physics model: $f_{\mathrm{P}}(T, \boldsymbol{z}_{\mathrm{P}}) := T_t - z_{\mathrm{P}} T_{ss}$.

**Latent variables**  By construction of $f_{\mathrm{P}}$, $z_{\mathrm{P}} \in \mathbb{R}$ is expected to work in the same manner as $a$ in the data-generating process. There was also $\boldsymbol{z}_{\mathrm{A}} \in \mathbb{R}^4$ in the full `NN+phys` and `NN+phys+reg` models. Meanwhile, we used $\boldsymbol{z}_{\mathrm{A}} \in \mathbb{R}^5$ (and no $\boldsymbol{z}_{\mathrm{P}}$) in the `NN-only` and `NN+solver` models.

**Decoder architecture**  We describe the decoder architecture of the full `NN+phys` and `NN+phys+reg` models. In $\mathcal{F}$, a PDE $f_{\mathrm{P}}(T, z_{\mathrm{P}}) + f_{\mathrm{A}} = 0$ was numerically solved with the finite difference method with the explicit scheme for length $\tau$ with temporal step size $\Delta t$. We modeled $f_{\mathrm{A}}$ with an MLP with two hidden layers of size 64. We used ELU with its[3] $\alpha = 1.0$ as activation function after the hidden layers. In the `NN-only` model, we modeled $f_{\mathrm{A}}$ with an MLP with a hidden layer of size 128.

**Encoder architecture**  We describe the encoder architecture of the full `NN+phys` and `NN+phys+reg` models. We modeled the recognition networks, $g_{\mathrm{A}}$ and $g_{\mathrm{P},2}$, with MLPs with five hidden layers of size 256, 256, 256, 64, and 32. We modeled $g_{\mathrm{P},1}(\boldsymbol{x}, \boldsymbol{z}_{\mathrm{A}})$ with an MLP with two hidden layers of size 256. We used ELU with its[3] $\alpha = 1.0$ as activation function after the hidden layers. We put a softplus function after the final output of $g_{\mathrm{P}}$ to make its output positive-valued.

---

[4] $\alpha$ and $\gamma$ here are different from the ones of the hyperparameters of the proposed regularizers.

**Replacement functions** To create the reduced model, we replaced $f_A$ by $h_A = 0$.

**Hyperparameters** We selected the hyperparameters, $\alpha$, $\beta$, and $\gamma$, from the following sets: $\alpha \in \{10^{-2}, 10^{-1}\}$, $\beta \in \{10^{-2}, 10^{-1}\}$, and $\gamma \in \{10^5, 10^6\}$. These ranges were chosen to roughly adjust the values of the corresponding regularizers to that of the ELBO. The configuration that achieved the best reconstruction error on the validation set was selected finally: $\alpha = 10^{-1}$, $\beta = 10^{-2}$, and $\gamma = 10^6$. In computing $R_{\mathrm{DA},2}$, we sampled $z_P^*$ from the uniform distribution on range $[0.005, 0.2]$.

**Optimization** We used the Adam optimizer with its[4] $\alpha = 10^{-3}$, $\gamma_1 = 0.9$, $\gamma_2 = 0.999$, and $\epsilon = 10^{-3}$. We ran iterations with mini-batch size 200 for 20000 epochs (i.e., 100,000 iterations in total) and saved the model that achieved the best validation reconstruction error.

### E.4 Galaxy images

**Data property** We used images of galaxies from a part of the Galaxy10 dataset[5]. We selected the 589 images of the "Disk, Edge-on, No Bulge" class to form an overall dataset. Each image is of size $69 \times 69$ with three channels, so $x_i \in \mathbb{R}^{69 \times 69 \times 3}$. We normalized the intensity values into range $[0, 1]$.

**Train/valid/test split** We separated the overall dataset them into training, validation, and test sets with 400, 100, and 89 images, respectively. In training, we performed data augmentation with random vertical/horizontal flips and random rotation, and thus the size of the training set was 8,000.

**Physics model** The physics model $f_P \colon \mathbb{R}^4 \to \mathbb{R}^{69 \times 69}$ is an exponential profile of the light distribution of galaxies whose input is $z_P := [I_0\ A\ B\ \vartheta]^\mathsf{T} \in \mathbb{R}^4_{>0}$, whose elements have the semantics introduced in the following. Let $[f_P]_{i,j}$ denote the $(i, j)$-element of the output of $f_P$. Then, for $1 \le i, j \le 69$,

$$[f_P]_{i,j} = I_0 \exp(-r_{i,j}),$$

where

$$r_{i,j}^2 := \frac{(\mathsf{X}_j \cos\vartheta - \mathsf{Y}_i \sin\vartheta)^2}{A^2} + \frac{(\mathsf{X}_j \sin\vartheta + \mathsf{Y}_i \cos\vartheta)^2}{B^2},$$

$$\mathsf{X}_j := 2 \cdot \frac{j-1}{68} - 1,$$

$$\mathsf{Y}_i := -2 \cdot \frac{i-1}{68} + 1.$$

$(\mathsf{X}_j, \mathsf{Y}_i)$ is the coordinate on the $69 \times 69$ even grid on $[-1, 1] \times [-1, 1]$. $I_0$ determines the overall magnitude of the light distribution, $A$ and $B$ determine the size of the ellipse of the light distribution, and $\vartheta$ determines its rotation. This model was used in a similar problem of Aragon-Calvo and Carvajal [4], where they only handle artificial images. See also, e.g., Erwin [20], for an extensive list of such light distribution models of galaxies.

**Latent variables** $z_P \in \mathbb{R}^4$ contains the information of intensity, semi-major and semi-minor axes, and rotation, as mentioned above. We used $z_A \in \mathbb{R}^2$ in the full `NN+phys` and `NN+phys+reg` models. Meanwhile, we used $z_A \in \mathbb{R}^6$ (and no $z_P$) in the `NN-only` model.

**Decoder architecture** There is no nontrivial equation-solving process this time because the physics model $f_P$ itself gives the closed-form solution. So the data-generating process in the full `NN+phys` and `NN+phys+reg` models is:

$$\mathcal{F}[f_P, f_{A,\mathrm{Unet}}, f_{A,\mathrm{tconv}}; z_P, z_A] := f_{A,\mathrm{Unet}}\big(f_P(z_P), f_{A,\mathrm{tconv}}(z_A)\big).$$

$f_{A,\mathrm{tconv}}$ is a neural net with transposed convolutional layers and given $z_A$, outputs a signal in $\mathbb{R}^{69 \times 69}$. $f_{A,\mathrm{Unet}}$ is a neural net with architecture similar to the U-Net, whose outputs are in $\mathbb{R}^{69 \times 69 \times 3}$. We used the rectified linear unit (ReLU) as activation function and applied batch normalization before each activation function. In the `NN-only` model, we modeled $f_A(z_A)$ only with a neural net with transposed convolutional layers whose output is in $\mathbb{R}^{69 \times 69 \times 3}$.

Note that we do not consider the `NN+solver` type of baseline as there appear no nontrivial solvers.

---

[5]The original images are from the Sloan Digital Sky Survey `www.sdss.org`, and the labels are from the Galaxy Zoo project `www.galaxyzoo.org`. The dataset is available a part of the `astroNN` package [41]

**Encoder architecture**  The architecture of $g_{P,2}$ and $g_A$ is similar to the one in Aragon-Calvo and Carvajal [4]. We put the softplus function after the final output of $g_P$ to make its output positive-valued. $g_{P,1}$ is simply $g_{P,1}(\boldsymbol{x}) := \sum_{i=1}^{3} c_i [\boldsymbol{x}]_i$, where $[\boldsymbol{x}]_i$ denotes the $i$-th channel of $\boldsymbol{x}$, and $c$'s are trainable parameters.

**Replacement functions**  To create the reduced model, we replaced $f_{A,\text{Unet}}$ by $h_A$ such that $h_A(\boldsymbol{\nu}) := [\boldsymbol{\nu}; \boldsymbol{\nu}; \boldsymbol{\nu}] \in \mathbb{R}^{69 \times 69 \times 3}$ (i.e., the repeat operator along the channel axis).

**Hyperparameters**  We selected the hyperparameter $\alpha$ from $\alpha \in \{10^{-2}, 10^{-1}, 1\}$. This range was chosen to roughly adjust the value of the corresponding regularizer to that of the ELBO. The others were fixed to be $\beta = 1$ and $\gamma = 10^3$; these values were also determined by roughly adjusting the order of the values of objectives. In computing $R_{\text{DA},2}$, we sampled from the uniform distributions on $I_0^* \in [0.5, 1]$, $A^* \in [0.1, 1.0]$, $e^* \in [0.2, 0.8]$, and $\vartheta^* \in [0, 3.142]$, where $B = A(1 - e)$.

### E.5  Human gait

**Physics model**  We modeled $f_P$ with a trainable Hamilton's equation as in [76, 24]:

$$f_P\left([\boldsymbol{p}^\mathsf{T} \quad \boldsymbol{q}^\mathsf{T}]^\mathsf{T}, \boldsymbol{z}_P\right) = \left[-\frac{\partial \mathcal{H}}{\partial \boldsymbol{q}}^\mathsf{T} \quad \frac{\partial \mathcal{H}}{\partial \boldsymbol{p}}^\mathsf{T}\right]^\mathsf{T},$$

where $\boldsymbol{p} \in \mathbb{R}^{d_\text{H}}$ is a generalized position, $\boldsymbol{q} \in \mathbb{R}^{d_\text{H}}$ is a generalized momentum, and $\mathcal{H} \colon \mathbb{R}^{d_\text{H}} \times \mathbb{R}^{d_\text{H}} \to \mathbb{R}$ is a Hamiltonian. We let $d_\text{H} = 3$ and modeled $\mathcal{H}$ with an MLP with two hidden layers of size.

**Latent variables**  $\boldsymbol{z}_P \in \mathbb{R}^{2d_\text{H}}$ is used as the initial condition of $\boldsymbol{p}$ and $\boldsymbol{q}$. There was also $\boldsymbol{z}_A \in \mathbb{R}^{15}$.

**Decoder architecture**  In the full NN+phys and NN+phys+reg models, the decoding process contains a numerical solver of ODE $f_P = 0$ with the Euler method. Its output is then transformed by $f_A$, an MLP with two hidden layers of size 512.

**Encoder architecture**  $g_P$ and $g_A$ are MLPs with five hidden layers of size 512, 512, 512, 64, 32.

**Replacement functions**  To create the reduced model, we replaced $f_A$ by an affine map $h_A$, where $h_A$ is applied to each snapshot of a sequence independently.

**Hyperparameters**  We selected the hyperparameter $\alpha$ from $\alpha \in \{10^{-3}, 10^{-2}, 10^{-1}, 1\}$. This range was chosen to roughly adjust the value of the corresponding regularizer to that of the ELBO. The other hyperparameters were just $\gamma = \beta = 0$ as we did not use the corresponding regularizers.

## F  Additional experimental results

We present additional experimental results including investigation of the sensitivity of hyperparameter values and some observation on training runtime.

### F.1  Forced damped pendulum

**Hyperparameter sensitivity**  We investigated the sensitivity of the performance with regard to the hyperparameters, i.e., the regularization coefficients, $\alpha$, $\beta$, and $\gamma$. We varied them around the nominal values, i.e., the setting with which the results were reported in the main text ($\alpha = 10^{-2}$, $\beta = 10^{-3}$, and $\gamma = 10^{-1}$; see also Appendix E). Figure F.1 summarizes the result. We can consistently observe the tendency that 1) NN+phys+reg is far better than phys-only in terms of the reconstruction error (upper row); and that 2) NN+phys+reg is far better than NN+phys in terms of the estimation error of physics parameter $\omega$ (lower row).

**Achieved hyperparameter values**  We examined the values of the regularizers for data augmentation. After training, $R_{\text{DA},1} \approx 0.5$ and $R_{\text{DA},2} \approx 2 \times 10^{-3}$ whereas $\|x\|_2^2 \approx 16$ on average. This result implies that the functionality of $g_{P,1}$ and $g_{P,2}$ are well controlled as intended.

**Training runtime**    In training, the `NN-only` model took about 5.13 seconds for 10 epochs, and the `NN+phys+reg` took about 10.9 seconds for 10 epochs, though we believe our implementation can still be improved for more efficiency. The difference probably stems from the physics-part encoder.

**More examples of reconstruction and extrapolation**    In the main text, we have shown only one example case of the reconstruction and extrapolation. In Figure F.2, we provide more examples on different test samples to facilitate further understanding of the result.

### F.2   Advection-diffusion system

**Hyperparameter sensitivity**    We investigated the sensitivity of the performance with regard to the hyperparameters $\alpha$, $\beta$, and $\gamma$. We varied these values around the nominal values, i.e., the setting with which the results were reported in the main text ($\alpha = 10^{-1}$, $\beta = 10^{-2}$, and $\gamma = 10^6$; see also hyperparameter settings in Appendix E). Figure F.3 summarizes the result. Across all the coefficient values, we can consistently observe the tendency similar to that in the pendulum data experiment.

**Achieved hyperparameter values**    We examined the values of the regularizers for data augmentation. After training, $R_{\mathrm{DA},1} \approx 0.01$ and $R_{\mathrm{DA},2} \approx 5 \times 10^{-7}$ whereas $\|x\|_2^2 \approx 458$ on average. This result implies that the functionality of $g_{\mathrm{P},1}$ and $g_{\mathrm{P},2}$ are well controlled as intended.

**Training runtime**    In training, the `NN-only` model took about 6.01 seconds for 10 epochs, and the `NN+phys+reg` took about 15.4 seconds for 10 epochs.

### F.3   Galaxy images

**Reconstruction**    In Figure F.4, we show examples of reconstruction of five test samples. While the `phys-only` model cannot recover the color information by construction, the other models that include neural nets reproduce the original colors to some extent. The reconstruction errors over the whole test set are reported in Table F.1. From these results, we can observe that the reconstruction performance is similar between `NN-only`, `NN+phys`, and `NN+phys+reg`. Despite the similar reconstruction performance, the `NN+phys+reg` model achieves clearly better generation performance as shown in the main text.

**Counterfactual generation**    In Figure F.5, we show the result of generation, where we varied the last element of $z_{\mathrm{P}}$ that corresponds to the angle of a galaxy in image, $\vartheta$. We examined the models trained without or with one of the regularizers, $R_{\mathrm{PPC}}$ (i.e., $\alpha = 0$); the other regularizers were always active. In Figure F.5, the case without the regularizer does not show reasonable generation with

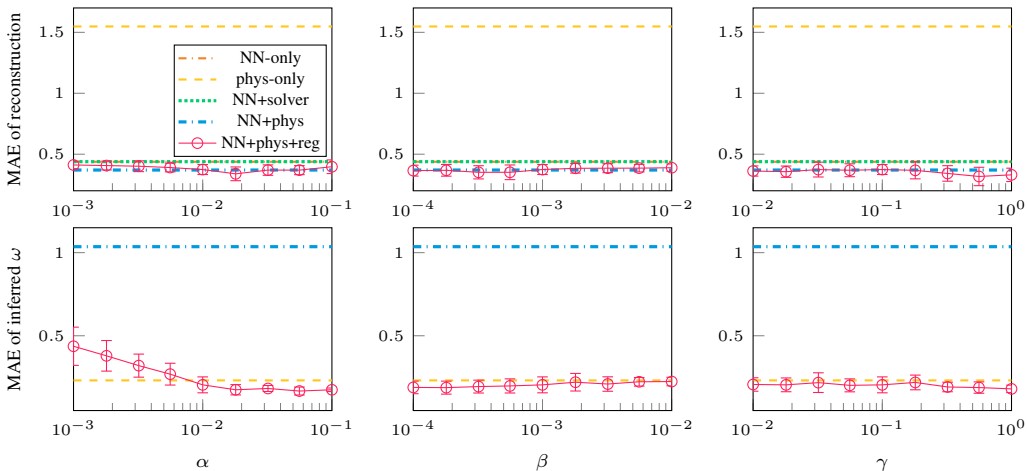

Figure F.1: Performances on the pendulum data with one of the hyperparameters ($\alpha$, $\beta$, or $\gamma$) varied around the nominal value, while the others maintained. Averages and SDs over five random trials are reported. Reference values are shown in dashed or dotted lines.

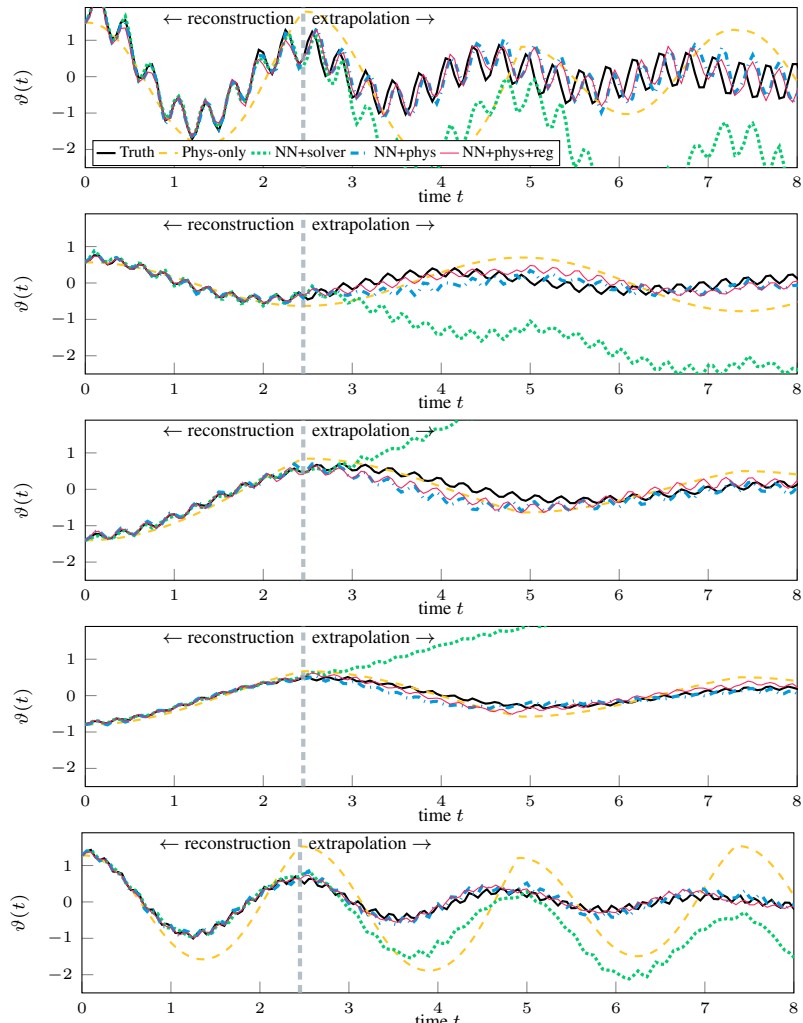

Figure F.2: Reconstruction and extrapolation of five test samples of the pendulum data. Range $0 \leq t < 2.5$ is reconstruction, whereas $t \geq 2.5$ is extrapolation. The bottom corresponds to the example presented in the main text.

Table F.1: Performances on test set of the galaxy image data. Averages (and SDs) over the whole test set are reported.

|  | MAE of reconstruction | |
| --- | --- | --- |
| NN-only | 0.0167 | $(3.0 \times 10^{-2})$ |
| Phys-only | 0.0264 | $(3.9 \times 10^{-2})$ |
| NN+phys(+reg), $\alpha = 0$ | 0.0188 | $(3.4 \times 10^{-2})$ |
| NN+phys+reg, $\alpha > 0$ | 0.0180 | $(3.3 \times 10^{-2})$ |

Table F.2: Performances on test set of the gait data. Averages (SDs) over 20 random trials are reported.

|  | MAE of reconstruction | |
| --- | --- | --- |
| Phys-only | 0.726 | $(1.0 \times 10^{-2})$ |
| NN+solver | 0.276 | $(1.5 \times 10^{-2})$ |
| NN+phys | 0.273 | $(9.0 \times 10^{-3})$ |
| NN+phys+reg | 0.259 | $(9.0 \times 10^{-3})$ |

different $\vartheta$. Note that $\vartheta < 0$ was never encountered during training as we set the range of the last element of $z_{\mathrm{P}}$ to be non-negative; nevertheless reasonable images are generated with $\vartheta < 0$.

**Latent variable** We computed the first two principal scores of $z_{\mathrm{A}}$ and plotted them with the corresponding image sample in Figure F.6. In the NN-only model, the distribution of $z_{\mathrm{A}}$ clearly

corresponds to the angle of the galaxy in images[6]. In contrast in the `NN+phys+reg` model, such a correspondence is not observed. This is a reasonable result because in `NN+phys+reg`, the semantic of galaxy angle is completely assigned to the last element of $z_P$.

### F.4 Human gait

**Reconstruction** The reconstruction errors over the whole test set are reported in Table F.2.

## G Extension

While the proposed framework is useful as shown in our experiments, there are several directions to go for possible technical improvement of the method. First, physics-integrated VAEs can be further combined with techniques to solve ODEs and PDEs with neural networks [62, 86, 85]. We supposed the use of differentiable numerical solvers if the model contains ODEs or PDEs, but such numerical solvers are often computationally heavy. Replacing them with neural net-based solutions will be useful for various applications. Second, while we defined the regularizer based on the (possibly loose) upper bound of KL divergence, we may use other dissimilarity measure of distributions or random variables, such as maximum mean discrepancy. Third, the proposed regularization method can be extended to other types of deep generative models; e.g., an extension to InfoVAE [94] is straightforward. Lastly, neural architecture search in the context of physics-integrated models [6] would be an interesting topic also in generative modeling.

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

---

[6]This might be a good property in some applications, but we do not want for it to happen in our `NN+phys+reg` model because the angle is rather manually encoded in an element of $z_P$, and $z_A$ should carry other information.

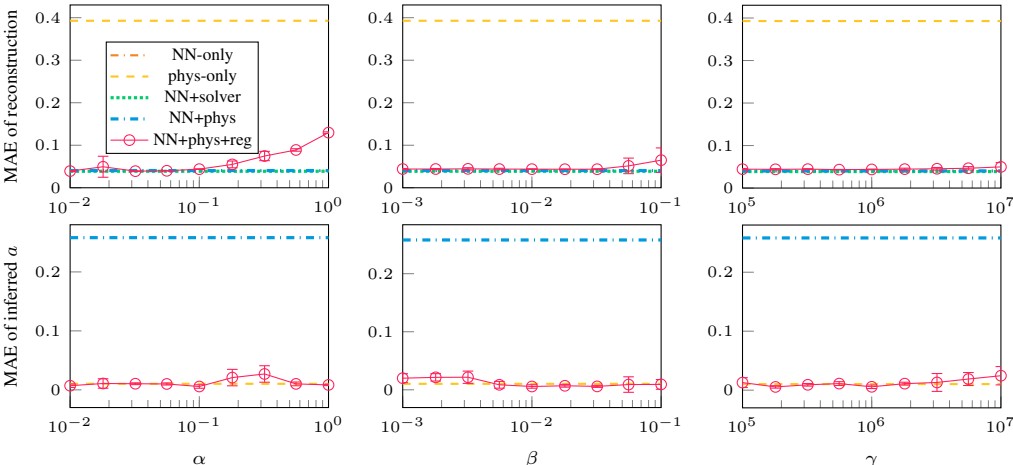

Figure F.3: Performances on the advection-diffusion data with one of the hyperparameters ($\alpha$, $\beta$, or $\gamma$) varied around the nominal value, while the others maintained. Averages and SDs over five random trials are reported. Reference values are shown in dashed or dotted lines.

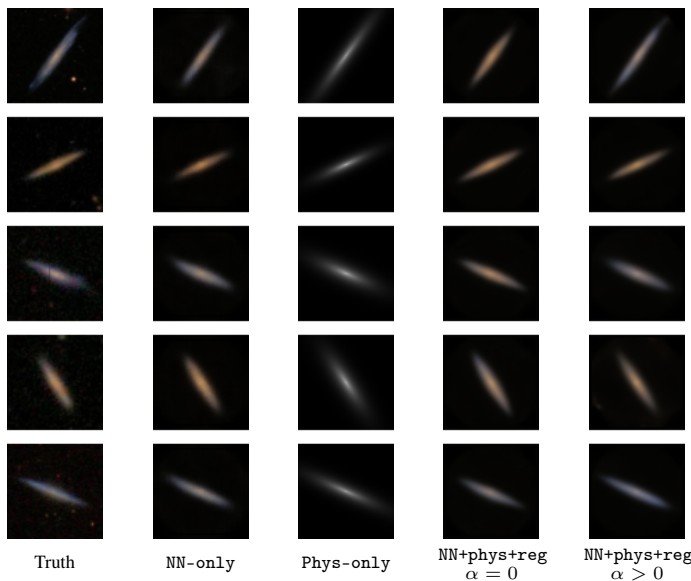

Truth  NN-only  Phys-only  NN+phys+reg $\alpha = 0$  NN+phys+reg $\alpha > 0$

Figure F.4: Reconstruction of five test samples of the galaxy images data. Best viewed in color.

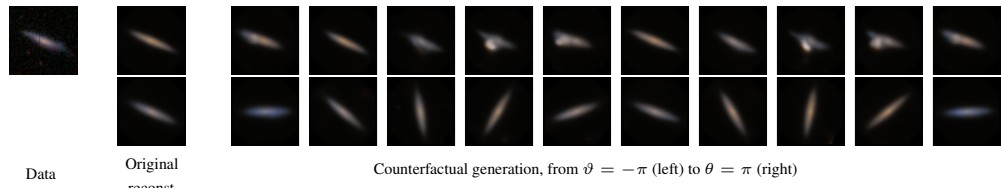

Data  Original reconst.  Counterfactual generation, from $\vartheta = -\pi$ (left) to $\theta = \pi$ (right)

Figure F.5: Counterfactual generation for the galaxy image data. (*1st column*) Original data sample. (*2nd column*) Original reconstruction of the sample. (*the rest*) Generation with varying $[z_P]_4$, which corresponds to the angle of galaxy in an image, $\vartheta$, from $-\pi$ to $\pi$. The upper row is with NN+phys(+reg) with $\alpha = 0$, and the lower row is with NN+phys+reg with $\alpha > 0$.

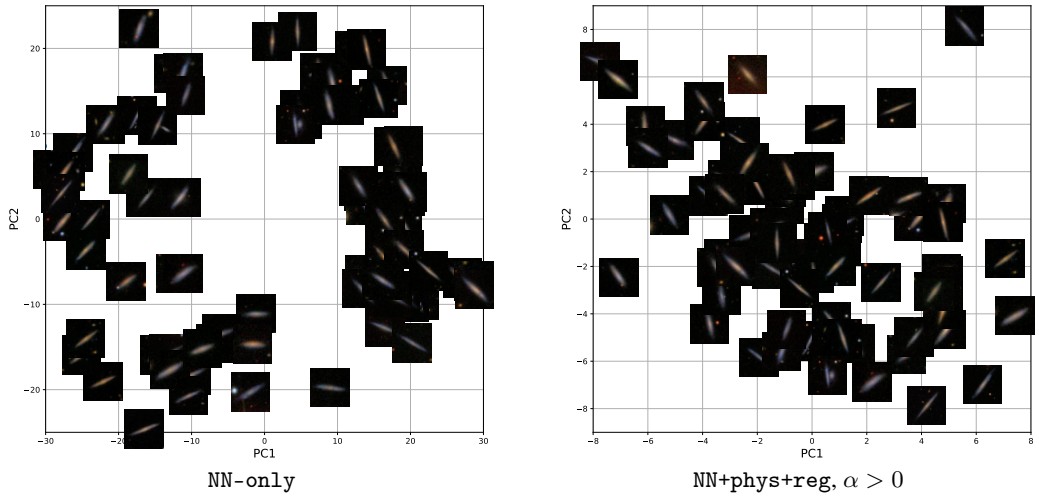

NN-only  NN+phys+reg, $\alpha > 0$

Figure F.6: Visualization of latent variable $z_A$ learned from the galaxy image data. The corresponding test data samples are shown at the points specified by the first two principal scores of $z_A$.

[4] M. A. Aragon-Calvo and J. C. Carvajal. Self-supervised learning with physics-aware neural networks – I. Galaxy model fitting. *Monthly Notices of the Royal Astronomical Society*, 498(3): 3713–3719, 2020.

[5] S. Ö. Arık, C.-L. Li, J. Yoon, R. Sinha, A. Epshteyn, L. T. Le, V. Menon, S. Singh, L. Zhang, N. Yoder, M. Nikoltchev, Y. Sonthalia, H. Nakhost, E. Kanal, and T. Pfister. Interpretable sequence learning for COVID-19 forecasting. arXiv:2008.00646, 2020.

[6] Y. Ba, G. Zhao, and A. Kadambi. Blending diverse physical priors with neural networks. arXiv:1910.00201, 2019.

[7] K. Beckh, S. Müller, M. Jakobs, V. Toborek, H. Tan, R. Fischer, P. Welke, S. Houben, and L. von Rueden. Explainable machine learning with prior knowledge: An overview. arXiv:2105.10172, 2021.

[8] A. Behjat, C. Zeng, R. Rai, I. Matei, D. Doermann, and S. Chowdhury. A physics-aware learning architecture with input transfer networks for predictive modeling. *Applied Soft Computing*, 96: 106665, 2020.

[9] F. d. A. Belbute-Peres, T. D. Economon, and J. Z. Kolter. Combining differentiable PDE solvers and graph neural networks for fluid flow prediction. In *Proceedings of the 37th International Conference on Machine Learning*, pages 2402–2411, 2020.

[10] G. Camps-Valls, D. H. Svendsen, J. Cortés-Andrés, Á. Moreno-Martínez, A. Pérez-Suay, J. Adsuara, I. Martín, M. Piles, J. Muñoz-Marí, and L. Martino. Living in the physics and machine learning interplay for earth observation. arXiv:2010.09031, 2020.

[11] F. P. Casale, A. Dalca, L. Saglietti, J. Listgarten, and N. Fusi. Gaussian process prior variational autoencoders. In *Advances in Neural Information Processing Systems 31*, pages 10369–10380, 2018.

[12] X. Chen, X. Xu, X. Liu, S. Pan, J. He, H. Y. Noh, L. Zhang, and P. Zhang. PGA: Physics guided and adaptive approach for mobile fine-grained air pollution estimation. In *Proceedings of the 2018 ACM International Joint Conference on Pervasive and Ubiquitous Computing and Wearable Computers*, pages 1321–1330, 2018.

[13] J. Chung, K. Kastner, L. Dinh, K. Goel, A. Courville, and Y. Bengio. A recurrent latent variable model for sequential data. In *Advances in Neural Information Processing Systems 28*, pages 2980–2988, 2015.

[14] K. Cranmer, J. Brehmer, and G. Louppe. The frontier of simulation-based inference. *Proceedings of the National Academy of Sciences*, page 201912789, 2020.

[15] M. Cranmer, S. Greydanus, S. Hoyer, P. Battaglia, D. Spergel, and S. Ho. Lagrangian neural networks. arXiv:2003.04630, 2020.

[16] E. de Bézenac, A. Pajot, and P. Gallinari. Deep learning for physical processes: Incorporating prior scientific knowledge. *Journal of Statistical Mechanics: Theory and Experiment*, 2019(12): 124009, 2019.

[17] W. De Groote, E. Kikken, E. Hostens, S. Van Hoecke, and G. Crevecoeur. Neural network augmented physics models for systems with partially unknown dynamics: Application to slider-crank mechanism. arXiv:1910.12212, 2019.

[18] M. Déchelle, J. Donà, K. Plessis-Fraissard, P. Gallinari, and M. Levy. Bridging dynamical models and deep networks to solve forward and inverse problems. NeurIPS workshop on Interpretable Inductive Biases and Physically Structured Learning, 2020.

[19] F. Djeumou, C. Neary, E. Goubault, S. Putot, and U. Topcu. Neural networks with physics-informed architectures and constraints for dynamical systems modeling. arXiv:2109.06407, 2021.

[20] P. Erwin. Imfit: A fast, flexible new program for astronomical image fitting. *The Astrophysical Journal*, 799(2):226, 2015.

[21] M. Fraccaro, S. K. Sønderby, U. Paquet, and O. Winther. Sequential neural models with stochastic layers. In *Advances in Neural Information Processing Systems 29*, pages 2199–2207, 2016.

[22] T. Frerix, D. Kochkov, J. A. Smith, D. Cremers, M. P. Brenner, and S. Hoyer. Variational data assimilation with a learned inverse observation operator. In *Proceedings of the 38th International Conference on Machine Learning*, pages 3449–3458, 2021.

[23] F. Golemo, P.-Y. Oudeyer, A. A. Taïga, and A. Courville. Sim-to-real transfer with neural-augmented robot simulation. In *Proceedings of the 2nd Conference on Robot Learning*, pages 817–828, 2018.

[24] S. Greydanus, M. Dzamba, and J. Yosinski. Hamiltonian neural networks. In *Advances in Neural Information Processing Systems 32*, pages 15379–15389, 2019.

[25] E. Heiden, D. Millard, E. Coumans, Y. Sheng, and G. S. Sukhatme. NeuralSim: Augmenting differentiable simulators with neural networks. arXiv:2011.04217, 2020.

[26] I. Higgins, L. Matthey, A. Pal, C. Burgess, X. Glorot, M. Botvinick, S. Mohamed, and A. Lerchner. $\beta$-VAE: Learning basic visual concepts with a constrained variational framework. In *Proceedings of the 5th International Conference on Learning Representations*, 2017.

[27] M. Jaques, M. Burke, and T. Hospedales. Physics-as-inverse-graphics: Unsupervised physical parameter estimation from video. In *Proceedings of the 8th International Conference on Learning Representations*, 2020.

[28] X. Jia, J. Willard, A. Karpatne, J. Read, J. Zwart, M. Steinbach, and V. Kumar. Physics guided RNNs for modeling dynamical systems: A case study in simulating lake temperature profiles. In *Proceedings of the 2019 SIAM International Conference on Data Mining*, pages 558–566, 2019.

[29] Y. Jiang, J. Sun, and C. K. Liu. Data-augmented contact model for rigid body simulation. arXiv:1803.04019, 2018.

[30] Y. Jiang, T. Zhang, D. Ho, Y. Bai, C. K. Liu, S. Levine, and J. Tan. SimGAN: Hybrid simulator identification for domain adaptation via adversarial reinforcement learning. arXiv:2101.06005, 2021.

[31] S. Kaltenbach and P.-S. Koutsourelakis. Physics-aware, probabilistic model order reduction with guaranteed stability. In *Proceedings of the 9th International Conference on Learning Representations*, 2021.

[32] M. Karl, M. Soelch, J. Bayer, and P. van der Smagt. Deep variational Bayes filters: Unsupervised learning of state space models from raw data. In *Proceedings of the 5th International Conference on Learning Representations*, 2017.

[33] G. E. Karniadakis, I. G. Kevrekidis, L. Lu, P. Perdikaris, S. Wang, and L. Yang. Physics-informed machine learning. *Nature Reviews Physics*, 2021.

[34] A. Karpatne, G. Atluri, J. Faghmous, M. Steinbach, A. Banerjee, A. Ganguly, S. Shekhar, N. Samatova, and V. Kumar. Theory-guided data science: A new paradigm for scientific discovery from data. *IEEE Transactions on Knowledge and Data Engineering*, 29(10):2318–2331, 2017.

[35] A. Karpatne, W. Watkins, J. Read, and V. Kumar. Physics-guided neural networks (PGNN): An application in lake temperature modeling. arXiv:1710.11431, 2017.

[36] S. Karra, B. Ahmmed, and M. K. Mudunuru. AdjointNet: Constraining machine learning models with physics-based codes. arXiv:2109.03956, 2021.

[37] R. G. Krishnan, U. Shalit, and D. Sontag. Structured inference networks for nonlinear state space models. In *Proceedings of the 31st AAAI Conference on Artificial Intelligence*, pages 2101–2109, 2017.

[38] F. Lanusse, P. Melchior, and F. Moolekamp. Hybrid physical-deep learning model for astronomical inverse problems. arXiv:1912.03980, 2019.

[39] V. Le Guen and N. Thome. Disentangling physical dynamics from unknown factors for unsupervised video prediction. In *Proceedings of the 2020 IEEE/CVF Conference on Computer Vision and Pattern Recognition*, pages 11471–11481, 2020.

[40] F. Leeb, Y. Annadani, S. Bauer, and B. Schölkopf. Structured representation learning using structural autoencoders and hybridization. arXiv:2006.07796, 2021.

[41] H. W. Leung and J. Bovy. Deep learning of multi-element abundances from high-resolution spectroscopic data. *Monthly Notices of the Royal Astronomical Society*, 483(3):3255–3277, 2018.

[42] L. Li, S. Hoyer, R. Pederson, R. Sun, E. D. Cubuk, P. Riley, and K. Burke. Kohn-Sham equations as regularizer: Building prior knowledge into machine-learned physics. *Phyiscal Review Letters*, 126(3):036401, 2020.

[43] Y. Li and S. Mandt. Disentangled sequential autoencoder. In *Proceedings of the 35th International Conference on Machine Learning*, pages 5670–5679, 2018.

[44] O. Linial, D. Eytan, and U. Shalit. Generative ODE modeling with known unknowns. arXiv:2003.10775, 2020.

[45] Y. Long and X. She. HybridNet: Integrating model-based and data-driven learning to predict evolution of dynamical systems. In *Proceedings of the 2nd Conference on Robot Learning*, pages 551–560, 2018.

[46] Z. Long, Y. Lu, X. Ma, and B. Dong. PDE-net: Learning PDEs from data. In *Proceedings of the 35th International Conference on Machine Learning*, pages 3208–3216, 2018.

[47] Z. Long, Y. Lu, and B. Dong. PDE-Net 2.0: Learning PDEs from data with a numeric-symbolic hybrid deep network. *Journal of Computational Physics*, 399:108925, 2019.

[48] M. Lutter, C. Ritter, and J. Peters. Deep Lagrangian networks: Using physics as model prior for deep learning. In *Proceedings of the 7th International Conference on Learning Representations*, 2019.

[49] I. Matei, J. de Kleer, C. Somarakis, R. Rai, and J. S. Baras. Interpretable machine learning models: A physics-based view. arXiv:2003.10025, 2020.

[50] V. Mehta, I. Char, W. Neiswanger, Y. Chung, A. O. Nelson, M. D. Boyer, E. Kolemen, and J. Schneider. Neural dynamical systems: Balancing structure and flexibility in physical prediction. arXiv:2006.12682, 2020.

[51] S. K. Mitusch, S. W. Funke, and M. Kuchta. Hybrid FEM-NN models: Combining artificial neural networks with the finite element method. *Journal of Computational Physics*, 446:110651, 2021.

[52] A. T. Mohan, N. Lubbers, D. Livescu, and M. Chertkov. Embedding hard physical constraints in neural network coarse-graining of 3D turbulence. arXiv:2002.00021, 2020.

[53] N. Muralidhar, J. Bu, Z. Cao, L. He, N. Ramakrishnan, D. Tafti, and A. Karpatne. PhyNet: Physics guided neural networks for particle drag force prediction in assembly. In *Proceedings of the 2020 SIAM International Conference on Data Mining*, pages 559–567, 2020.

[54] H. V. Nguyen and T. Bui-Thanh. Model-constrained deep learning approaches for inverse problems. arXiv:2105.12033, 2021.

[55] A. Nutkiewicz, Z. Yang, and R. K. Jain. Data-driven Urban Energy Simulation (DUE-S): A framework for integrating engineering simulation and machine learning methods in a multi-scale urban energy modeling workflow. *Applied Energy*, 225:1176–1189, 2018.

[56] S. Pakravan, P. A. Mistani, M. A. Aragon-Calvo, and F. Gibou. Solving inverse-PDE problems with physics-aware neural networks. arXiv:2001.03608, 2020.

[57] S. Pawar, O. San, B. Aksoylu, A. Rasheed, and T. Kvamsdal. Physics guided machine learning using simplified theories. arXiv:2012.13343, 2020.

[58] D. Pitchforth, T. Rogers, U. Tygesen, and E. Cross. Grey-box models for wave loading prediction. *Mechanical Systems and Signal Processing*, 159:107741, 2021.

[59] D. C. Psichogios and L. H. Ungar. A hybrid neural network-first principles approach to process modeling. *AIChE Journal*, 38(10):1499–1511, 1992.

[60] Z. Qian, W. R. Zame, L. M. Fleuren, P. Elbers, and M. van der Schaar. Integrating expert ODEs into Neural ODEs: Pharmacology and disease progression. arXiv:2106.02875, 2021.

[61] M. Raissi. Deep hidden physics models: Deep learning of nonlinear partial differential equations. *Journal of Machine Learning Research*, 19(25):1–24, 2018.

[62] M. Raissi, P. Perdikaris, and G. E. Karniadakis. Physics-informed neural networks: A deep learning framework for solving forward and inverse problems involving nonlinear partial differential equations. *Journal of Computational Physics*, 378:686–707, 2019.

[63] M. Reichstein, G. Camps-Valls, B. Stevens, M. Jung, J. Denzler, N. Carvalhais, and Prabhat. Deep learning and process understanding for data-driven Earth system science. *Nature*, 566 (7743):195–204, 2019.

[64] R. Reinhart, Z. Shareef, and J. Steil. Hybrid analytical and data-driven modeling for feed-forward robot control. *Sensors*, 17(2):311, 2017.

[65] H. Ren, R. Stewart, J. Song, V. Kuleshov, and S. Ermon. Learning with weak supervision from physics and data-driven constraints. *AI Magazine*, 39(1):27–38, 2018.

[66] R. Rico-Martínez, J. S. Anderson, and I. G. Kevrekidis. Continuous-time nonlinear signal processing: A neural network based approach for gray box identification. In *Proceedings of the IEEE Workshop on Neural Networks for Signal Processing*, pages 596–605, 1994.

[67] M. A. Roehrl, T. A. Runkler, V. Brandtstetter, M. Tokic, and S. Obermayer. Modeling system dynamics with physics-informed neural networks based on Lagrangian mechanics. arXiv:2005.14617, 2020.

[68] S. Saemundsson, A. Terenin, K. Hofmann, and M. Deisenroth. Variational integrator networks for physically structured embeddings. In *Proceedings of the 23rd International Conference on Artificial Intelligence and Statistics*, pages 3078–3087, 2020.

[69] B. Schölkopf, F. Locatello, S. Bauer, N. R. Ke, N. Kalchbrenner, A. Goyal, and Y. Bengio. Towards causal representation learning. arXiv:2102.11107, 2021.

[70] U. Sengupta, M. Amos, J. S. Hosking, C. E. Rasmussen, M. Juniper, and P. J. Young. Ensembling geophysical models with Bayesian neural networks. In *Advances in Neural Information Processing Systems 33*, 2020.

[71] N. Shlezinger, J. Whang, Y. C. Eldar, and A. G. Dimakis. Model-based deep learning. arXiv:2012.08405, 2020.

[72] S. K. Singh, R. Yang, A. Behjat, R. Rai, S. Chowdhury, and I. Matei. PI-LSTM: Physics-infused long short-term memory metwork. In *Proceedings of the 18th IEEE International Conference on Machine Learning and Applications*, pages 34–41, 2019.

[73] X. Sun, T. Xue, S. M. Rusinkiewicz, and R. P. Adams. Amortized synthesis of constrained configurations using a differentiable surrogate. arXiv:2106.09019, 2021.

[74] D. J. Tait and T. Damoulas. Variational autoencoding of PDE inverse problems. arXiv:2006.15641, 2020.

[75] M. L. Thompson and M. A. Kramer. Modeling chemical processes using prior knowledge and neural networks. *AIChE Journal*, 40(8):1328–1340, 1994.

[76] P. Toth, D. J. Rezende, A. Jaegle, S. Racanière, A. Botev, and I. Higgins. Hamiltonian generative networks. In *Proceedings of the 8th International Conference on Learning Representations*, 2020.

[77] K. Um, R. Brand, Y. R. Fei, P. Holl, and N. Therey. Solver-in-the-Loop: Learning from differentiable physics to interact with iterative PDE-Solvers. In *Advances in Neural Information Processing Systems 33*, pages 6111–6122, 2020.

[78] F. A. Viana, R. G. Nascimento, A. Dourado, and Y. A. Yucesan. Estimating model inadequacy in ordinary differential equations with physics-informed neural networks. *Computers & Structures*, 245:106458, 2021.

[79] L. von Rueden, S. Mayer, K. Beckh, B. Georgiev, S. Giesselbach, R. Heese, B. Kirsch, J. Pfrommer, A. Pick, R. Ramamurthy, M. Walczak, J. Garcke, C. Bauckhage, and J. Schuecker. Informed machine learning – A taxonomy and survey of integrating knowledge into learning systems. arXiv:1903.12394v2, 2020.

[80] L. von Rueden, S. Mayer, R. Sifa, C. Bauckhage, and J. Garcke. Combining machine learning and simulation to a hybrid modelling approach: Current and future directions. In *Advances in Intelligent Data Analysis XVIII*, number 12080 in Lecture Notes in Computer Science, pages 548–560. 2020.

[81] Z. Y. Wan, P. Vlachas, P. Koumoutsakos, and T. Sapsis. Data-assisted reduced-order modeling of extreme events in complex dynamical systems. *PLOS ONE*, 13(5):e0197704, 2018.

[82] Q. Wang, F. Li, Y. Tang, and Y. Xu. Integrating model-driven and data-driven methods for power system frequency stability assessment and control. *IEEE Transactions on Power Systems*, 34(6):4557–4568, 2019.

[83] S. Wang, Y. Teng, and P. Perdikaris. Understanding and mitigating gradient flow pathologies in physics-informed neural networks. *SIAM Journal on Scientific Computing*, 43(5):A3055–A3081, 2021.

[84] J. Willard, X. Jia, S. Xu, M. Steinbach, and V. Kumar. Integrating physics-based modeling with machine learning: A survey. arXiv:2003.04919, 2020.

[85] L. Yang, X. Meng, and G. E. Karniadakis. B-PINNs: Bayesian physics-informed neural networks for forward and inverse PDE problems with noisy data. *Journal of Computational Physics*, 425:109913, 2021.

[86] Y. Yang and P. Perdikaris. Physics-informed deep generative models. arXiv:1812.03511, 2018.

[87] Ç. Yıldız, M. Heinonen, and H. Lähdesmäki. ODE2VAE: Deep generative second order ODEs with Bayesian neural networks. In *Advances in Neural Information Processing Systems 32*, pages 13412–13421, 2019.

[88] Y. Yin, V. Le Guen, J. Dona, I. Ayed, E. de Bézenac, N. Thome, and P. Gallinari. Augmenting physical models with deep networks for complex dynamics forecasting. In *Proceedings of the 9th International Conference on Learning Representations*, 2021.

[89] C.-C. Young, W.-C. Liu, and M.-C. Wu. A physically based and machine learning hybrid approach for accurate rainfall-runoff modeling during extreme typhoon events. *Applied Soft Computing*, 53:205–216, 2017.

[90] A. Zeng, S. Song, J. Lee, A. Rodriguez, and T. Funkhouser. TossingBot: Learning to throw arbitrary objects with residual physics. In *Proceedings of Robotics: Science and Systems*, 2019.

[91] R. Zhang, Y. Liu, and H. Sun. Physics-guided convolutional neural network (PhyCNN) for data-driven seismic response modeling. *Engineering Structures*, 215:110704, 2020.

[92] Z. Zhang, R. Rai, S. Chowdhury, and D. Doermann. MIDPhyNet: Memorized infusion of decomposed physics in neural networks to model dynamic systems. *Neurocomputing*, 428: 116–129, 2021.

[93] S. Zhao, J. Song, and S. Ermon. The information autoencoding family: A Lagrangian perspective on latent variable generative models. In *Proceedings of the 34th Conference on Uncertainty in Artificial Intelligence*, 2018.

[94] S. Zhao, J. Song, and S. Ermon. InfoVAE: Balancing learning and inference in variational autoencoders. In *Proceedings of the 33rd AAAI Conference on Artificial Intelligence*, pages 5885–5892, 2019.