# OpenReview forum: "Physics-Integrated Variational Autoencoders for Robust and Interpretable Generative Modeling"
_NeurIPS.cc/2021/Conference — NeurIPS 2021 Poster_

### Official Review · Reviewer_c9CR · 2021-07-14

**Rating:** 4
**Confidence:** 4

**Summary:**

This paper describes a VAE architecture in which some latent variables are semantically related to our prior knowledge of the physical model. For instance, this represents the (unknown) angular velocity or diffusion coefficient. The semantic is defined the
functional form of  the decoder part. Then the objective function is augmented with regularization terms to better control how the model will take into account the physical model and the prior knowledge. The experimental part relies on two synthetic datasets and two reals ones.

**Main Review:**

The interaction between Data and theory-driven modeling is clearly an important topic.  The introduction addresses the topic in maybe a too broad perspective. The authors want to keep their work as general as possible, but you can think about a lot of possibilities depending on the task or the scientific area (model, optimization, generation, generalization, robustness, ... ). The paper could be easier to follow with an example of purpose.

The example provides in section 2.1 really helps to understand the purpose. Maybe it could appear in the introduction (you should avoid the confusing use of theta, NNet parameters for the decoder in 2.2.2, and part of the observation in 2.1 and 5.1). Then the real purpose becomes clearer: some latent variables are semantically related to our prior knowledge of the physical model. For instance, this represents the (unknown) angular velocity or diffusion coefficient. The semantic is defined the functional form of  the decoder.

The contributions of the paper are clearly interesting. However, there is a for me a clarity issue. The notations (with theta) are sometime misleading. The section 3 describes some nice ideas but it is really difficult to follow.  For instance the section "on grounding physics encoder by physics-based data augmentation"  describes a nice idea but would it be possible to simplify a bit the discourse to go a bit more straight to the point.

Finally the experimental part (section 5) does not really provide a meaningful insight. Maybe, the galaxy part could be discarded to spare place for the others. Moreover, the results on advection-diffusion system are not so easy to interpret. They are a bit different from the pendulum one.




**Time Spent Reviewing:**

1

---

> ### Author Response · Authors · 2021-08-08
> **Authors' Response to Reviewer c9CR**
>
> Dear Reviewer c9CR,
>
>
> Thank you very much for reviewing. We appreciate the suggestion about the structure of the introduction.
>
> ### (I)
>
> > The section 3 describes some nice ideas but it is really difficult to follow. For instance the section "on grounding physics encoder by physics-based data augmentation" describes a nice idea but would it be possible to simplify a bit the discourse to go a bit more straight to the point.
>
> In the physics-based data augmentation in Section 3.2, we want to use a physics model for self-supervising the hybrid generative model. The key technical challenge here is that because the physics model may be incomplete, the artificial signals generated from it and the real signals may have different natures. To compensate such difference, we arrange the particular functionality of the physics encoder $g_P=g_{P,2} \circ g_{P,1}$, such that $g_{P,1}$ computes a "physics-only" version of real signals, and then $g_{P,2}$ computes the physics latent variable. In this way, $g_{P,2}$ can be directly self-supervised by the physics model because it is supposed to receive only "physics-only" signals. To this end, a couple of regularizers are introduced.
>
> We would like to start Section 3.2 with a comprehensive explanation like above, whereas it is now scattered over the section. Moreover, we gave further clarification about Section 3.2 in Answer (XIV) to Reviewer LEU9.
>
> ### (II)
>
> > Finally the experimental part (section 5) does not really provide a meaningful insight.
>
> We did not really understand why the reviewer thought that the experimental section did not provide meaningful insight. In the series of experiments, we showed the reasonable performance of the proposed model in terms of *both* reconstruction and inference (in Table 1), the extrapolation capability (in Figures 1 and 3), a kind of conditional generation with manipulating physics latent variable (in Figure 2), the unconditional generation (in Figure 4), and the intuitive investigation of the behavior of the hybrid model with and without the proposed regularizers (in Figure 5). Moreover, in the appendix, we examined the (non)sensitivity of the performance to the hyperparameters (in Figures F.1 and F.2) and conducted the visual investigation of the learned latent variables (in Figure F.5). We would appreciate it if the reviewer could comment more on what else should be presented to show the validity and the effectiveness of the proposed method.
>
> ### (III)
>
> > Moreover, the results on advection-diffusion system are not so easy to interpret. They are a bit different from the pendulum one.
>
> While we are not sure what prevented the reviewer from interpreting the results, we would like to kindly refer to Answer (V) to Reviewer LEU9, which would probably clarify the concern. In a nutshell, the main message of Table 1 is that the proposed method (phys+NN+reg) achieves *reasonable* performance in *both* reconstruction and inference. The selected hyperparameter in the advection-diffusion experiment was probably not the optimal one, which does not diminish the main message.

---

> ### Author Response · Authors · 2021-09-02
> **Initial concern clarified?**
>
> Thank you again for reviewing our paper. In the initial review, the clarity of some descriptions and the meaningfulness of the experiments were questioned. We were not sure which points were particularly questioned and responded with our best guess. However, we have not received further feedback and would like to hear if our response could address the initial concerns well. We really appreciate the reviewer's effort in the reviewing and discussion process.

---

### Official Review · Reviewer_EtH5 · 2021-07-15

**Rating:** 7
**Confidence:** 4

**Summary:**

This paper develops a general approach to integrating physical domain knowledge into the latent space, encoder, and decoder of VAEs. The latent space is separate into a physics component and learned component, where the former is grounded to some unobserved physical quantity or system variable. The decoder is a composition of a physics-based component with known functional form (e.g. an ODE of the latent physical variable) and a neural component which also takes the learned part of the latent space as input. The encoder is a composition of two stages, the first of which extracts the learned latents, and the second of which recovers the physically grounded latent variables. In order to train this model to encourage interpretability and usage of the physics components, the authors develop several regularization terms which are added to the ELBO loss, thereby constrain the power of highly flexible neural encoders and decoders, ensuring that the physical components play a meaningful role in explaining the data. The importance of such regularization (and the value of physics components) is demonstrated on a series of physical system identification tasks.

**Limitations And Societal Impact:**

Ideally the authors would discuss both the limitations and societal impact more -- I have highlighted some potential limitations in my review above. Regarding societal impact, maybe the authors could mention how physically grounded models would aid better scientific discovery, but also lead to more accurate / robust surveillance and prediction systems (e.g. in the case of identifying and predicting human gait).

**Main Review:**

This is a nice paper which generalizes existing approaches that integrate physical domain knowledge within partially specified ODEs (e.g. Yin et al, ICRL 2021), while allowing for the specification of priors over latent system variables, as in the VAE framework. To my knowledge, this paper is original and valuable in the following ways:
1. It generalizes integration of physical domain knowledge into ML models beyond the additive combination of learned components with physics ODEs.
2. It applies physics integration in the context of Bayesian generative modeling, allowing for a range of additional tasks like unsupervised learning, counterfactual simulation / sample generation, and Bayesian parameter estimation.
2. It shows how to overcome the well-known problem of VAE decoders and latent spaces being overly flexible (and hence ignoring expert or physical guidance through priors and expert-specified equations) through regularization.

The experiments also appear to be sufficiently thorough in demonstrating both the value of integrating physical knowledge and the importance of regularization when learning the behavior of physical systems like diffusion-advection, galaxy image generation, and human gait prediction. For these reasons I recommend acceptance of the paper.

I just have a few questions / points of clarification. Regarding the proposed regularization terms, I wonder if there are Bayesian interpretations for them, especially given the reference to posterior predictive checks? For example, would it be possible to derive something like $R_{PPC}$ by imposing a prior on the form of the decoder, similar to how the prior on the latent space acts as a regularizer in variational inference?

I also wonder whether the $R_{PPC}$ method is going to work well in the case where the observations are a pretty complex / high dimensional function of the physical variables, e.g. images of humans walking from the joint velocities in the case of human gait. In that case, while the full decoder should be able to do reasonable job of matching the observed data, it seems like the physics-only decoder would struggle, unless the function $h_A$ were a learned complex function like a CNN as well.

In this sort of situation then, it seems like having the $R_{PPC}$ term would actually hurt learning, because the physics-only decoder would do a very bad job of fitting the data even after training / conditioning, and the full decoder would be regularized towards being similar. This could be fixed if modelers had enough domain knowledge to specify $h_A$ (e.g. $h_A$ could be a differentiable renderer in the case of generating images of humans walking), but it seems like this won't always be the case. It would be good if the authors could discuss this situation in the final manuscript.

Finally, I was a bit confused by the use of the term "harness" to mean something like "constrain" or "regularize", because "harness" also has the meaning of "utilize / exploit / make use of". It might be a good idea to replace "harness" with "regularize" or "constrain" in phrases like "harness trainable components".

== Post-Rebuttal ==

Thank you to the authors for their detailed responses and clarifications. I believe that they have adequately responded to the questions raised by myself and other reviewers, and I will maintain my score of 7.

**Time Spent Reviewing:**

4

---

> ### Author Response · Authors · 2021-08-08
> **Authors' Response to Reviewer EtH5**
>
> Dear Reviewer EtH5,
>
> Thank you very much for reviewing. We appreciate raising interesting discussions.
>
> ### (I)
>
> > Regarding the proposed regularization terms, I wonder if there are Bayesian interpretations for them, especially given the reference to posterior predictive checks? For example, would it be possible to derive something like $R_{PPC}$ by imposing a prior on the form of the decoder, similar to how the prior on the latent space acts as a regularizer in variational inference?
>
> This is an interesting viewpoint that we have not considered. In general, model checking including PPC is a step of Bayesian inference that is usually regarded to be distinct from the other steps of Bayesian inference, i.e., model construction and posterior inference [9, Section 6.1]. Hence, it would be natural to think that our regularizer $R_\text{PPC}$ is not technically the same as imposing a prior on the decoder (= a kind of model construction). In fact, a prior on the decoder, if any, would be integrated out when computing posterior predictive distributions, whereas it would directly appear in a prior-induced regularization term. Our idea is to use such a model checking-like method not as model checking but as regularization during training, unlike traditional Bayesian inference.
>
> Nonetheless, giving different interpretations of $R_\text{PPC}$ or similar regularizer would be a very important topic of future studies as it may provide a unified view on controlling machine learning models. One of possibly related methodologies is the posterior regularization [Ganchev+ 2010], in which they consider a generalized Bayesian inference with soft constraints on posteriors.
>
> Ganchev et al., Posterior Regularization for Structured Latent Variable Models, JMLR 11:2001-2049, 2010.
>
> ### (II)
>
> > I also wonder whether the $R_{PPC}$ method is going to work well in the case where the observations are a pretty complex / high dimensional function of the physical variables, e.g. images of humans walking from the joint velocities in the case of human gait. In that case, while the full decoder should be able to do reasonable job of matching the observed data, it seems like the physics-only decoder would struggle, unless the function $h_A$ were a learned complex function like a CNN as well.
>
> > In this sort of situation then, it seems like having the $R_{PPC}$ term would actually hurt learning, because the physics-only decoder would do a very bad job of fitting the data even after training / conditioning, and the full decoder would be regularized towards being similar. This could be fixed if modelers had enough domain knowledge to specify $h_A$ (e.g. $h_A$ could be a differentiable renderer in the case of generating images of humans walking), but it seems like this won't always be the case. It would be good if the authors could discuss this situation in the final manuscript.
>
> We agree this is an important discussion when applying the model to highly complex datasets, while it is basically beyond the scope of the current paper as generating overly high-dimensional signals like videos is itself a very challenging problem. Actually, a kind of “hurting learning” phenomenon probably due to the $R_\text{PPC}$ term can be observed in our advection-diffusion experiment of Table 1, where the reconstruction error of the NN+phys+reg model is slightly larger than those of some baselines. We elaborated this issue in Answer (V) to Reviewer LEU9.
>
> We would like to mention possible strategies as follows:
>
> - A naive yet efficient way would be, as you suggested, to handcraft a semantically-grounded observation function such as a differentiable render.
> - Another possible way is to learn an observation function, i.e., a function from physics’ state space (e.g., human joints angles) to an observation space (e.g., videos), separately from a generative model. This is often possible if we know the state space, and even if we do not know the exact dynamics there. For example, suppose that we want to learn a physics-integrated generative model of human walking videos. In order to learn an observation function from the joint angles to videos, we may use some labeled datasets for human pose estimation from videos. A point here is that the datasets for learning the observation function can be different from the dataset for learning the generative model; we can use larger datasets for the observation function, e.g., datasets including non-walking human poses. Once such an observation function is (roughly) learned, then in learning the generative model, we can fine-tune it with a small learning rate.
> - Apart from the goodness of observation function, a straightforward way to alleviate the detrimental effect of $R_\text{PPC}$ is to make its coefficient $\alpha$ small, but it may not always be good.
> - It is also possible to use different statistical distances for defining $R_\text{PPC}$. We simply used the KL divergence in the current paper. When observation noise is Gaussian, minimizing the KL needs to minimize the geometric distance between the decoders’ outputs, which may hurt the learning. Hence, instead of the KL, it may be a good idea to define $R_\text{PPC}$ with dependence-based statistical distances such as the mutual information and the Hilbert-Schmidt independence criterion.
>
> ### (III)
>
> > Finally, I was a bit confused by the use of the term "harness" to mean something like "constrain" or "regularize", because "harness" also has the meaning of "utilize / exploit / make use of". It might be a good idea to replace "harness" with "regularize" or "constrain" in phrases like "harness trainable components".
>
> Thank you for pointing out this. We will modify the word usage.

---

### Official Review · Reviewer_LEU9 · 2021-07-17

**Rating:** 6
**Confidence:** 3

**Summary:**

The authors propose to incorporate physics models into variational autoencoder (VAE) architectures in a principled manner. They design regularization terms in the ELBO loss to achieve a VAE with a latent space that is partially grounded in interpretable physics elements. The experiments demonstrate that the resulting VAE performs well in reconstruction error, is able to infer physically interpretable parameters in its latent space, and generalizes beyond the provided data.

**Limitations And Societal Impact:**

As mentioned above, the authors have not adequately addressed the limitations of their specific method in the paper (besides general limitations with respect to incorporating complicated physics models into gray-box models). Discussion of the drop in reconstruction performance in Table 1 (advection-diffusion experiments) would be a good candidate for the discussion of limitations. Furthermore, I would encourage the authors to think about potential inadvertent societal consequences of the proposed method such as any concerns with deployment of neural network architectures in safety critical systems.

**Main Review:**

There has recently been a lot of work in the space of neural networks grounded in physics. The authors provide a reasonable literature review in the main paper, and further contextualization in existing work in the appendix. Since I do not work directly in this space, it is difficult for me to judge the exact novelty of the proposed methods. That being said, the approach seems to be effective, and the authors do their due diligence in extensive experimentation and comparison to methods similar in motivation. I would recommend the authors further expand their literature review to include works in the space of physics-grounded learning for the purpose of control. I provide some examples below.

[1] Gupta, Jayesh K., et al. "Structured mechanical models for robot learning and control." Learning for Dynamics and Control. 2020.

[2] Richards, S. M., et al. "Adaptive-Control-Oriented Meta-Learning for Nonlinear Systems." Robotics science and systems. 2021.

The ideas in the submission are evaluated extensively across multiple experimental domains. The experiments strongly support the claims made in the paper in terms of reconstruction quality, latent space interpretability, and the flexibility of the VAE to handle non-additive physics-based and learned components. The general structure of the baselines considered poses an ablation study where each of the ablations resemble a related work in the space. However, some of the baselines (e.g., phys-only) seem a bit contrived for a number of the experimental settings. For example, in the case of the pendulum and the advection-diffusion system, the phys-only baseline has no hope of performing well as the authors only provide it with partial information about the simulated physics. Granted, this is not the case for the galaxy and human gait experiments. While only qualitative results are shown for the physics-only model in the main paper, quantitative results are reported in the appendix for these experiments and the trends are consistent with the main paper. It would have been good to provide a bit more intuition for why and how the proposed model is able to outperform the baselines (for example in lines 277-281).

One question I had while the reading the paper is how does the VAE ensure that the learned $f_{A}$ component is not zero since all of the regularization terms appear to focus on driving the VAE closer to the physics model. Further experimental analysis demonstrating the performance of the learned $f_{A}$ term would be helpful.

My biggest concern with the paper is its lack of discussion of the limitations of the proposed method. The authors do refer to a general limitation in the space of physics-based learning in the difficulty of integrating complex physics models. However, no specific limitations of the proposed approach are discussed. Furthermore, there appears to be a lack of transparency in the discussion of the results for the advection-diffusion experiments. Table 1 reports that the proposed method underperforms the considered baselines in terms of reconstruction ability. This on its own is not grounds for rejection as there are extensive experiments demonstrating the proposed method's value. However, the discussion in the paper does not address this limitation, but rather ignores it (lines 309-310). The authors should discuss their hypothesis for why this drop in relative performance was observed for this experiment.

I have a few additional clarification questions with respect to the experiments:

1. Why were the results for the NN+phys (with no regularization) baseline not displayed in Fig. 1?

2. In Fig. 2, were both the $z_{P}$ and the $\omega$ values changed for each of the plots?

3. For the pendulum experiments, why was the $f_{A,2}$ term introduced? What is the purpose of the latent variable $z_{A,2}$ beyond the function of $z_{A,1}$?

4. In Table 1, why were standard deviations reported instead of standard error?

Although the paper is very well written, the notation made it difficult to follow at times. I may have missed this definition earlier in the paper, but is $f$ defined prior to line 97 which is where it appears to be first mentioned? Line 106 was difficult to follow notationally. The space of $S_{f_{p}}$ is not specified. I would at first assume that $S_{f_{P}}$ is in $Z_{P}$, but in the second half of the statement, it serves as the sole argument of $f_{A}$. This is confusing since the domain of $f_{A}$ is set to $Y_{P} \times Z_{A}$.

The most difficult section to read was Sec. 3.2, which is critical to the understanding of the paper. I had to read this section over multiple times and I am still not sure I am following the reasoning. First, I do not understand how $\mathcal{F}$ is a functional but it is able to take a value output from $h_{A}$ instead of a function as an argument (line 181). What is the difference between $x^*$ and $x^*_{z^*_{P}}$? Lines 187-194 were particularly confusing and difficult to understand. There is a lot of recursive reasoning and notation here. For example, $x'$ should resemble $x^*$ which is the physics-only reconstruction from the original data $x$ and latent code $z_{A}$. In Eq. (9), the output of $g_{P,1}$ is made to be close to $x'$, which is in turn computed using $g_{P,1}$. In Eq. (10), the input to $g_{P,2}$ is $x^*_{z^*_{P}}$ instead of $x'$, which is meant to provide self-supervision for $x'$ to be close to $x^*_{z^*_{P}}$. Simplifying the notation and streamlining the logic in this section would help the reader understand the proposed method better.

I found a couple minor typos in the paper (see below) but otherwise the polish of the paper was very good.

1. Typo in line 71: 'One one' should read 'On one'.

2. Line 207: 'Ones...are one' is awkward grammatically.

3. Typo in line 95: 'solve' should be 'solves'.

4. The references should be proofread (e.g., to ensure the year is not entered twice in a citation).

Overall, the authors propose an interesting and empirically successful method for physics-based learning using VAEs. Their approach demonstrates good generalizability and encourages a partially interpretable latent space. With the additional of a transparent discussion of the limitations of the method and improved notational clarity, this work will be a good contribution!

**Time Spent Reviewing:**

5

---

> ### Author Response · Authors · 2021-08-08
> **Authors' Response to Reviewer LEU9 (1/2)**
>
> Dear Reviewer LEU9,
>
> Thank you very much for reviewing. We appreciate the constructive feedback.
>
> ### (I)
>
> > However, some of the baselines (e.g., phys-only) seem a bit contrived for a number of the experimental settings. For example, in the case of the pendulum and the advection-diffusion system, the phys-only baseline has no hope of performing well as the authors only provide it with partial information about the simulated physics.
>
> While this view is correct, we would like to note that in Table 1, the phys-only baseline is informative as it provides reference values of the inference error; the NN+phys+reg model should achieve inference errors similar or better compared to those of the phys-only baseline and indeed do so. In contrast, the NN+phys baseline results in significantly worse inference errors.
>
> ### (II)
>
> > It would have been good to provide a bit more intuition for why and how the proposed model is able to outperform the baselines (for example in lines 277-281).
>
> The advantage of the physics model integration can be explained from multiple perspectives. In terms of the generalization capability (i.e., loss values on test data), the presence of a physics model enables us to specify a good hypothesis space, with which learning efficiency can improve. In terms of generation, the presence of semantically grounded latent variables is obviously advantageous as it enables us to understand and control the generation process. In terms of extrapolation, a physics-based part of a hybrid model is in general robust to distribution shifts, and thus it allows more natural extrapolation than a purely data-driven model. We are happy to add such intuitive explanations in the paper, while rigorous analyses will be interesting for future studies.
>
> ### (III)
>
> > One question I had while the reading the paper is how does the VAE ensure that the learned $f_A$ component is not zero since all of the regularization terms appear to focus on driving the VAE closer to the physics model.
>
> We do not have a mechanism to ensure non-zero $f_A$ because $f_A$ should be close to zero or identity when appropriate. For example, if a given physics model happens to be very faithful to given data, $f_A$ should be minimally working. We assume that a physics model may be incomplete but always be trustable (i.e., does not include irrelevant terms), and thus neural nets $f_A$ should be suppressed as long as ELBO (or other objectives) is not affected too much.
>
> Meanwhile, considering a mechanism to ensure non-zero $f_A$ will also be an interesting topic. For example, if one has a belief that a physics model likely contains some irrelevant terms, they should be compensated by neural nets. In such a case, the contribution of $f_A$ should be kept non-zero. A future study may deal with such an advanced setting.
>
> ### (IV)
>
> > My biggest concern with the paper is its lack of discussion of the limitations of the proposed method. The authors do refer to a general limitation in the space of physics-based learning in the difficulty of integrating complex physics models. However, no specific limitations of the proposed approach are discussed.
>
> Thank you for the suggestion. One of the specific limitations of the method would be the additional effort needed to tune the hyperparameters $\alpha$, $\beta$, and $\gamma$. We empirically found that the overall performance was not really sensitive to a small difference of these values (see Figures F.1 and F.2 in the appendix). Nonetheless, finding appropriate orders of magnitude certainly adds some workload especially when the data/model is large. Moreover, finding the very best values of the hyperparameters that achieve the optimal validation loss would be more laborious, though it would not be always necessary in practice.
>
> This limitation is related to the issue that the regularizers may deteriorate the other objectives (i.e., ELBO or reconstruction errors) as pointed out. Indeed, the value of $\alpha$ selected in the advection-diffusion experiment of Table 1 was probably not the best one. We discuss this issue below.
>
> ### (V)
>
> > Furthermore, there appears to be a lack of transparency in the discussion of the results for the advection-diffusion experiments. Table 1 reports that the proposed method underperforms the considered baselines in terms of reconstruction ability. This on its own is not grounds for rejection as there are extensive experiments demonstrating the proposed method's value. However, the discussion in the paper does not address this limitation, but rather ignores it (lines 309-310). The authors should discuss their hypothesis for why this drop in relative performance was observed for this experiment.
>
> We really appreciate your precise reading that raised such a detailed discussion. The proposed regularizers can likely inhibit optimal fitting of the other objectives as mentioned above, which is not surprising because they limit the flexibility of the neural nets.
>
> Actually, the results of the NN+phys+reg model shown in Table 1 are not necessarily with the best hyperparameters. In the advection-diffusion experiment of Table 1, we only tried $2^3$ combinations of $\alpha\in$ {$0.01,0.1$}, $\beta\in$ {$0.01,0.1$}, and $\gamma\in$ {$10^5,10^6$} and selected the one with the best validation performance ($\alpha=0.1$, $\beta=0.01$, $\gamma=10^6$). However, seemingly the selected configuration was not really optimal, and probably a smaller $\alpha$ would have been better — we tried the case of $\alpha=0.032$, $\beta=0.01$, $\gamma=10^6$ in the sensitivity experiment of Figure F.2, in which the average reconstruction error of the NN+phys+reg model was $0.0390$ (with $\text{SD}=4.5 \times 10^{-4}$). This value is quite comparable to those of the baselines in Table 1. We also note that in this case, the inference error was $0.0103$ (with $\text{SD}=1.5 \times 10^{-3}$), which is also similar to the values in Table 1.
>
> We did not report these better values in Table 1 for the following reasons. First, we wanted to align the condition of the two experiments in Table 1 (i.e., pendulum and advection-diffusion) in the sense that the hyperparameters were tuned only roughly. Second, we do not think reporting the very best values is necessary there. The main message of Table 1 is that the NN+phys+reg model can achieve *reasonable* performance both in reconstruction and in inference, unlike the phys-only baseline (that completely fails reconstruction) and the NN+phys baseline (that completely fails inference). Here we say that performance is *reasonable* if it is far from the worst cases (i.e., phys-only for reconstruction / NN+phys for inference). Hence, the exact order of the merit in Table 1 was not the main issue there. Nonetheless, it would be a good example of the proposed method’s specific limitation as discussed above, and we would like to add some remarks about it.
>
> ### (VI)
>
> > Why were the results for the NN+phys (with no regularization) baseline not displayed in Fig. 1?
>
> In the particular sample of Figure 1, the difference of the NN+phys+reg model from the NN+phys baseline was smaller than the differences from the other baselines. We did not want to mess up the figure in the limited space and thus omitted it. For some other samples, however, these two behaved in more obviously different manners, and we are willing to add new larger figures also with such examples in the appendix.
>
> ### (VII)
>
> > In Fig. 2, were both the $z_P$ and the $\omega$ values changed for each of the plots?
>
> Yes, in the following sense: $z_P$ was changed for generating the signals with the learned models (dotted blue and solid red lines), and $\omega$ was changed for generating the true signals with the true data-generating process (solid black line). A point is that the true signals with the changed $\omega$ were not used to compute the latent variables of the generated signals. Once $z_A$ and $z_P$ are computed from the original signal shown in the center panel of Figure 2, the generated signals in the other panels were computed with $z_A$ maintained and $z_P$ changed. The true signals were created just for comparison.
>
> ### (VIII)
>
> > For the pendulum experiments, why was the $f_{A,2}$ term introduced? What is the purpose of the latent variable $z_{A,2}$ beyond the function of $z_{A,1}$?
>
> In the pendulum experiment, we used not only $f_{A,1}$ (inside ODE) but also $f_{A,2}$ (outside ODE) because using both achieved better fitting. Only with $f_{A,1}$, fitting was not impossible, but the loss decrease was slow probably because fitting the external force term with $\cos$ was inefficient only with the medium-sized MLP as $f_{A,1}$. We tried larger MLPs as $f_{A,1}$, which was okay but inevitably resulted in slow training due to the large NN inside ODE. So, it was just a result of an empirical model design.
>
> ### (IX)
>
> > In Table 1, why were standard deviations reported instead of standard error?
>
> In general, the standard error (SE) describes the uncertainty of an estimation of the population mean, whereas the sample standard deviation (SD) is an estimation of the population SD. In our experiments, the performance indices actually fluctuate due to randomness (i.e., different random seeds), and we want to know the extent of such fluctuation = population SDs. Thus, we think that reporting sample SDs is natural as it provides a direct estimation of the population SD. If one wants to know how certain the mean performances are different (i.e., statistical significance), SEs would be more natural to see. Meanwhile, since we can directly estimate SEs from SDs if the sample size is known, reporting SDs and sample sizes would be reasonable enough in such a situation, too.
>
> **(continues to Authors' Response to Reviewer LEU9 (2/2))**

---

> ### Author Response · Authors · 2021-08-08
> **Authors' Response to Reviewer LEU9 (2/2)**
>
> **(continued from Authors' Response to Reviewer LEU9 (1/2))**
>
> ### (X)
>
> > Although the paper is very well written, the notation made it difficult to follow at times. I may have missed this definition earlier in the paper, but $f$ is defined prior to line 97 which is where it appears to be first mentioned?
>
>  We are sorry for the confusion. At Line 97, we mean by $f$ both $f_A$ and $f_B$. We modify the description.
>
> ### (XI)
>
> > Line 106 was difficult to follow notationally. The space of $S_{f_p}$ is not specified. I would at first assume that $S_{f_P}$ is in $Z_P$, but in the second half of the statement, it serves as the sole argument of $f_A$. This is confusing since the domain of $f_A$ is set to $Y_P \times Z_A$.
>
> In the case where a physics model has a closed-form solution $S_{f_P}$, we can think that the space of $S_{f_P}$ coincides with $Y_P$. The other argument $z_A \in Z_A$ was mistakenly dropped at the end of Line 106. It should have been $\mathcal{F}[f_P,f_A] := f_A(S_{f_P}, z_A)$.
>
> ### (XII)
>
> > First, I do not understand how $\mathcal{F}$ is a functional but it is able to take a value output from $h_A$ instead of a function as an argument (line 181).
>
> You can think that $\mathcal{F}$ is a functional:
>
> - that takes the composition of functions, $f_A \circ f_P$, originally in Eq. (3); and
> - that takes another composition of functions, $h_A \circ f_P$, later at Line 181.
>
> In either way, $\mathcal{F}$ does evaluation(s) of the function that it takes as an argument.
>
> It is natural to suppose that $\mathcal{F}$ is a functional because it may include some equation-solving process as mentioned at the beginning of Section 2.2.2. For example, $\mathcal{F}[f]$ may be an ODE solver of an ODE $f=0$, which needs access to the full function $f$ instead of a finite number of pointwise values of $f$, in order to conduct the numerical integration from arbitrary initial conditions. Note that $f$ here may be $f_A \circ f_P$ or $h_A \circ f_P$.
>
> ### (XIII)
>
> > What is the difference between $x^*$ and $x^*_{z^*_P}$?
>
> $x^*$ is the same with $x^*_{z^*_P}$; the subscript was just omitted. Sorry again for the confusing omission. We will improve it.
>
> ### (XIV)
>
> > Lines 187-194 were particularly confusing and difficult to understand. There is a lot of recursive reasoning and notation here. For example, $x'$ should resemble $x^*$ which is the physics-only reconstruction from the original data $x$ and latent code $z_A$. In Eq. (9), the output of $g_{P,1}$ is made to be close to $x'$, which is in turn computed using $g_{P,1}$. In Eq. (10), the input to $g_{P,2}$ is $x^*_{z_P^*}$ instead of $x'$, which is meant to provide self-supervision for $x'$ to be close to $x^*_{z_P^*}$. Simplifying the notation and streamlining the logic in this section would help the reader understand the proposed method better.
>
> While your understanding seems to be mostly correct, the last point "to provide self-supervision for $x'$ to be close to $x^*_{z_P^*}$" sounds a little odd. Note that $x'$ does not appear in Eq. (10). The purpose of Eq. (10) is rather to provide self-supervision for $g_{P,2}(x^*_{z_P^*})$ to be close to $z_P^*$. In other words, it self-supervises $g_{P,2}$ to correctly infer the true physics latent code $z_P^*$ given the signal generated from it, $x^*_{z_P^*}$.
>
> We agree that the reasoning sounds recursive. Nonetheless, we would say such recursive reasoning is inevitable to some extent because of the inherent recursiveness of the process. In this sense, the $R_{\text{DA},1}$ in Eq. (9) *per se* would not make much sense — it is meaningful only if the remaining part of the physics encoder, $g_{P,2}$, is working as desired, i.e., $g_{P,2}$ computes the physics latent code $z_P$ given "physics-only" signals. The $R_{\text{DA},2}$ in Eq. (10) works to ensure such desirable functionality of $g_{P,2}$, as mentioned above.
>
> For simplifying the notation, we will use a unified representation for $x^*_{z_P^*}$ and $x'$ because their definitions are similar. While dramatically simplifying the logic would be difficult due to the inherent recursiveness of the process, we will clarify the flow of information between the quantities, such as $z_P^*$, $x^*_{z_P^*}$, $x'$, and the outputs of $g_{P,1}$ and $g_{P,2}$, using a small diagram.

---

> ### Comment · Reviewer_LEU9 · 2021-08-30
> **Keeping my original score**
>
> Thank you to the authors for the clarifications! Including these points of discussion and clarifications in the paper would help the clarity of the submission. Overall, I am going to keep my score in leaning towards an accept.

---

> > ### Author Response · Authors · 2021-09-03
> > **Thank you for the feedback**
> >
> > Thank you very much for checking our response. We are glad to hear it helped the clarity. Meanwhile, we are wondering if you still feel some concerns that keep the rating to be 6 (and not more). In the initial review, you commented as follows:
> >
> > > Overall, the authors propose an interesting and empirically successful method for physics-based learning using VAEs. Their approach demonstrates good generalizability and encourages a partially interpretable latent space. With the additional of a transparent discussion of the limitations of the method and improved notational clarity, this work will be a good contribution!
> >
> > We believe we addressed the additions you suggested as follows: a transparent discussion of the limitation in our Answers (IV) and (V) and improvement of notation in our Answers (X), (XI), (XIII), and (XIV). We would really appreciate it if you could tell remaining concerns on these points if any, in order to further improve the paper. If you found the concerns well addressed, it would be wonderful if the rating also could be kindly updated accordingly.
> >
> > Thank you again for your engagement in the discussion.

---

### Decision · Program_Chairs · 2021-09-27

**Decision:**

Accept (Poster)

**Comment:**

The paper proposes to integrate a physics-based model within the Variational Auto-Encoder framework. Two of the reviewers are in favor of the paper and point out that the paper possesses many interesting features:
- The paper generalizes the integration of physical domain knowledge into ML models beyond the additive combination of learned components with physics ODEs.
- It applies physics integration in the context of deep generative modeling.
- It shows how to overcome the well-known problem of VAE decoders and latent spaces being overly flexible through regularization.
- The ideas in the paper are evaluated extensively across multiple experimental domains.

The third reviewer is inclided towards the rejection of the paper. However, the reviewer did not take part in the discussion and spent only an hour on the paper.
Overall, I find the points raised by the reviewers properly addressed by the authors in the rebuttal. Similarly, two reviewers seem to be satisfied with the rebuttal. Therefore, I am in favor of accepting the paper.